# Chromatin accessibility underlies synthetic lethality of SWI/SNF subunits in ARID1A-mutant cancers

Timothy W R Kelso[1], Devin K Porter[1], Maria Luisa Amaral[2], Maxim N Shokhirev[2], Christopher Benner[3], Diana C Hargreaves[1]*

[1]Molecular and Cell Biology Laboratory, Salk Institute for Biological Studies, California, United States; [2]The Razavi Newman Integrative Genomics and Bioinformatics Core Facility, Salk Institute for Biological Studies, California, United States; [3]Department of Medicine, University of California San Diego, California, United States

**Abstract** ARID1A, a subunit of the SWI/SNF chromatin remodeling complex, is frequently mutated in cancer. Deficiency in its homolog ARID1B is synthetically lethal with ARID1A mutation. However, the functional relationship between these homologs has not been explored. Here, we use ATAC-seq, genome-wide histone modification mapping, and expression analysis to examine colorectal cancer cells lacking one or both ARID proteins. We find that ARID1A has a dominant role in maintaining chromatin accessibility at enhancers, while the contribution of ARID1B is evident only in the context of ARID1A mutation. Changes in accessibility are predictive of changes in expression and correlate with loss of H3K4me and H3K27ac marks, nucleosome spacing, and transcription factor binding, particularly at growth pathway genes including *MET*. We find that ARID1B knockdown in ARID1A mutant ovarian cancer cells causes similar loss of enhancer architecture, suggesting that this is a conserved function underlying the synthetic lethality between ARID1A and ARID1B.
DOI: https://doi.org/10.7554/eLife.30506.001

*For correspondence: dhargreaves@salk.edu

Competing interests: The authors declare that no competing interests exist.

## Introduction

Mutations in ARID1A have recently been identified in diverse cancer types, including ovarian, endometrial, and colorectal cancer (*Kadoch et al., 2013*). The ARID1A protein is the largest subunit of the multi-protein SWI/SNF chromatin remodeling complex, with two mammalian homologs, ARID1B and ARID2, also found in this complex; ARID1A and ARID1B associate with the BAF complex, and ARID2 with the PBAF complex (*Wang et al., 1996*). Interestingly, compared to ARID1A, mutations in ARID1B and ARID2 are rare in cancer, apart from in neuroblastoma and hepatocellular carcinoma (*Fujimoto et al., 2012*; *Sausen et al., 2013*). ARID1A and ARID1B are often co-expressed and are mutually exclusive in the SWI/SNF complex (*Wang et al., 2004*), with each containing an ARID domain considered to facilitate non-specific DNA binding (*Chandler et al., 2013*; *Wilsker et al., 2004*). ARID1B is synthetic lethal with ARID1A mutation in cancer cell lines and fibroblasts, consistent with the presence of intact ARID1B-containing complexes (*Helming et al., 2014*; *Mathur et al., 2017*). The reason for the selectivity for ARID1A mutations in cancer is not understood, and the functional basis for the synthetic lethal relationship between ARID1A and ARID1B has not yet been determined.

SWI/SNF chromatin remodeling complexes disrupt DNA-histone contacts to rearrange the nucleosome landscape, utilizing energy from ATP hydrolysis to remodel nucleosomes in vitro (*Côté et al., 1994*; *Imbalzano et al., 1994*; *Kwon et al., 1994*; *Owen-Hughes et al., 1996*; *Pazin et al., 1994*;

*Whitehouse et al., 1999*) and in vivo (*Bao et al., 2015*; *Bossen et al., 2015*; *John et al., 2008*; *Takaku et al., 2016*; *Weinmann et al., 1999*). Bound extensively across the genome at promoters and enhancers, these complexes can both activate and repress gene expression (*Ho et al., 2009*), and ARID proteins can have cooperative and antagonistic roles in expression, with frequent instances of co-binding at promoters and enhancer elements (*Raab et al., 2015*).

A recent study by Mathur and colleagues characterized SWI/SNF genomic occupancy in HCT116 colorectal carcinoma cells with wild-type ARID1A or homozygous truncating ARID1A mutations (*Mathur et al., 2017*). Despite widespread binding of SWI/SNF to promoters and enhancers, removal of ARID1A caused specific loss of the complex at enhancers of developmental genes, with the H3K27ac mark of active enhancers diminished at these sites. Similarly, in demonstrating that loss of ARID1A can initiate neoplastic transformation in non-tumorigenic cells, Lakshminarasimhan and colleagues identified a small number of changes in gene expression, which correlated with alterations in histone modifications but no significant change in accessibility at transcription start sites (TSSs) or CTCF insulator sites (*Lakshminarasimhan et al., 2017*). To date, the in vivo role of ARID proteins in chromatin remodeling at a genome-wide level has not been directly explored.

To study this question, we made use of isogenic wild-type and ARID1A homozygous mutant HCT116 lines. We further subjected these lines to ARID1B knockdown to study the role of ARID1B in the ARID1A wild-type and mutant context. We performed the Assay for Transposase Accessible Chromatin with high-throughput sequencing (ATAC-seq) to assess genome-wide changes in DNA accessibility, which provides a direct measure of chromatin remodeling activity by SWI/SNF complexes (*Buenrostro et al., 2013*). We find that loss of ARID1A in wild-type HCT116 cells results in dramatic changes in chromatin accessibility, while ARID1B knockdown has no effect. In contrast, ARID1B knockdown in ARID1A mutant cells results in further up or down-regulation of accessibility at ARID1A-dependent and unique sites. Regions sensitive to ARID1A or ARID1B loss are predominantly found at enhancers and distal regulatory sites, where ARID1A and ARID1B are required for the maintenance of active enhancer histone marks. Additionally, ARID1A and ARID1B are critical for the binding of AP-1 family members at enhancers, consistent with diminished nucleosome spacing around AP-1 motifs upon loss of ARID1A and ARID1B. These alterations are highly correlated with changes in gene expression upon loss of one or both ARID proteins and we observe significant changes in the expression of genes encoding signaling intermediates in cell growth and adhesion, including *MET*. Applying similar techniques in a naturally occurring ARID1A-mutant ovarian cancer cell line, we find that knockdown of ARID1B results in loss of accessibility and active histone marks around AP-1 motifs at enhancers, suggesting a common mechanistic function for ARID proteins across cancer types. Our results demonstrate a role for ARID1A and ARID1B in maintaining chromatin accessibility and define nucleosome remodeling as a critical function underlying the tumor suppressor role of ARID1A and the synthetic lethal relationship of ARID1A and ARID1B.

## Results

### ARID1A and ARID1B have unique functional roles in the maintenance of chromatin accessibility

We employed isogenic HCT116 colorectal carcinoma cells that are wild-type for ARID1A (WT) or engineered for homozygous ARID1A protein loss (ARID1A$^{-/-}$) by introduction of an early stop codon (Q456*) by gene trap (Horizon Discovery). We further infected these lines with vectors for shRNA knockdown of ARID1B (ARID1B KD) or a scrambled control (WT or ARID1A$^{-/-}$). We performed two independent lentiviral infections as biological replicates to control for viral integration. As expected, ARID1A protein was abundantly expressed in WT HCT116 cells and completely absent in ARID1A$^{-/-}$ HCT116 cells (*Figure 1—figure supplement 1A*). Knockdown of ARID1B resulted in a 70–90% reduction in normal protein levels in both WT and ARID1A$^{-/-}$ HCT116 cells compared to a scrambled control (*Figure 1—figure supplement 1B*). We analyzed chromatin accessibility in these cells using ATAC-seq (*Buenrostro et al., 2013*). Loss of ARID1A in WT HCT116 cells dramatically altered overall chromatin accessibility, resulting in thousands of increased and decreased sites, while ARID1B KD surprisingly had no effect (*Figure 1A*). In contrast, ARID1B KD in ARID1A$^{-/-}$ HCT116 cells resulted in hundreds of changed sites, primarily at regions where accessibility was lost (*Figure 1A*). These results are consistent with the synthetic lethal relationship observed for ARID1A and ARID1B in

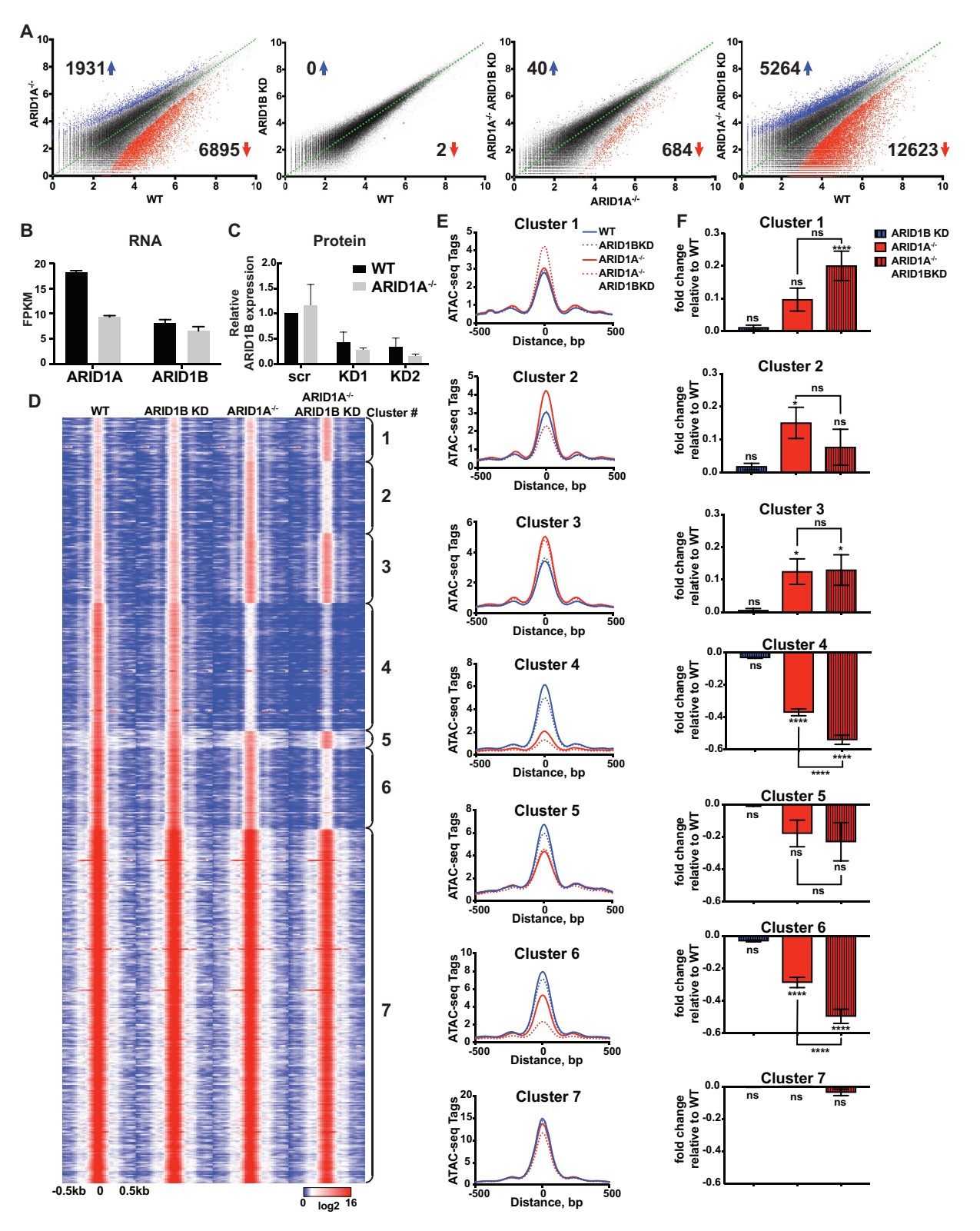

**Figure 1.** ARID1A and ARID1B have unique and shared roles in the maintenance of chromatin accessibility. (**A**) Differential peak calls from ATAC-seq in HCT116 WT cells expressing shRNAs to scrambled control (WT) or ARID1B (ARID1B KD), or HCT116 ARID1A$^{-/-}$ cells expressing shRNAs to scrambled control (ARID1A$^{-/-}$) or ARID1B (ARID1A$^{-/-}$ ARID1B KD). These cells were validated for the expression, complex association, and function of ARID1A and ARID1B (*Figure 1—figure supplement 1A–D*). Blue and red dots represent differential ATAC-seq peaks whose read density increased or decreased by

*Figure 1 continued on next page*

*Figure 1 continued*

1.5 fold or more (FDR < 0.05), respectively, in two independent biological replicates. Numbers in the upper left and lower right corners refer to numbers of called peaks increased or decreased in accessibility. Percent of changed sites was calculated from total number of accessible sites (*Figure 1—figure supplement 2A*). Replicate experiments showed strong correlation (*Figure 1—figure supplement 2B*) and overlap with DNAseI Hypersensitivity data (*Figure 1—figure supplement 2C*). (B) FPKM expression of ARID1A and ARID1B quantified by RNA-seq of two biological replicates from wild-type (WT) and ARID1A$^{-/-}$ (ARID1A$^{-/-}$) HCT116 cells. (C) ARID1B protein expression in wild-type (WT) and ARID1A$^{-/-}$ (ARID1A$^{-/-}$) HCT116 cells expressing shRNAs to scrambled control (scr) or ARID1B (KD1, KD2). ARID1B levels are quantified relative to TBP control (*Figure 1—figure supplement 1B*). Values shown are averages from three biological replicates where error bars indicate standard deviation. (D) *k*-means clustering analysis of ATAC-seq read density (log2) from WT, ARID1B KD, ARID1A$^{-/-}$, and ARID1A$^{-/-}$ ARID1B KD HCT116 cells. Reads are centered on the middle of the accessible peak ±0.5 kb. (E) Average ATAC-seq read density in WT, ARID1B KD, ARID1A$^{-/-}$, and ARID1A$^{-/-}$ ARID1B KD HCT116 cells centered on accessible peaks from Clusters 1–7. (F) Fold change in RNA expression relative to WT HCT116 cells for the top 25% of expressed genes nearest to called accessible peaks from Clusters 1–7. Ns, not significant; *p<0.05; **p<0.01; ***p<0.001, ****p<0.0001. Gene Set Enrichment analysis of hallmark gene sets was performed on annotated sites from Clusters 1–7 (*Figure 1—figure supplement 3*).

DOI: https://doi.org/10.7554/eLife.30506.002

The following source data and figure supplements are available for figure 1:

**Source data 1.** ARID1B protein expression in WT and ARID1A$^{-/-}$ HCT116 cells with ARID1B knockdown.
DOI: https://doi.org/10.7554/eLife.30506.006

**Source data 2.** Proliferation of WT and ARID1A$^{-/-}$ HCT116 cells with ARID1B knockdown.
DOI: https://doi.org/10.7554/eLife.30506.007

**Figure supplement 1.** Knockdown of ARID1B is synthetically lethal with ARID1A deletion in HCT116 cells.
DOI: https://doi.org/10.7554/eLife.30506.003

**Figure supplement 2.** Quality control of ATAC-seq datasets.
DOI: https://doi.org/10.7554/eLife.30506.004

**Figure supplement 3.** Hierarchical clustering of enrichment p-values (log$_e$) using Gene Set Enrichment Analysis (GSEA) with Hallmark gene sets.
DOI: https://doi.org/10.7554/eLife.30506.005

cancer cell proliferation (*Helming et al., 2014*; *Mathur et al., 2017*). Indeed, ARID1B loss had no effect on WT HCT116 cells but strongly repressed proliferation of ARID1A$^{-/-}$ HCT116 cells (*Figure 1—figure supplement 1C*). Overall, deficiency in both ARID1A and ARID1B led to decreased accessibility at 12,623 sites (12.5%) and increased accessibility at 5264 (5.2%) of 101,140 total accessible sites (*Figure 1A*, *Figure 1—figure supplement 2A*). We observed high correlation between biological replicates for all cell genotypes (*Figure 1—figure supplement 2B*) and our ATAC-seq accessibility patterns were consistent with the DNAseI hypersensitive profile of HCT116 cells from the ENCODE project (*Figure 1—figure supplement 2C*). Semi-quantitative measurements revealed that ARID1A is approximately 7-fold more abundant than ARID1B in WT HCT116 cells (data not shown). However, loss of ARID1A did not affect ARID1B transcript levels (*Figure 1B*) or protein abundance by western blot (*Figure 1C*), indicating that the effect of ARID1B KD in ARID1A$^{-/-}$ HCT116 cells is not due to compensatory upregulation of ARID1B. Our results suggest that loss of ARID1A and ARID1B predominantly represses accessibility at thousands of sites, implying an important role for these factors in maintaining open chromatin.

To explore how accessible sites are controlled by different ARID subunits, we clustered peaks from WT, ARID1A$^{-/-}$ and ARID1A$^{-/-}$ ARID1B KD HCT116 cells based on peak width and ARID1A dependence (*Figure 1D*). This generated 7 clusters that displayed distinct changes in accessibility caused by loss of ARID1A and ARID1B (*Figure 1E*). As shown, ARID1B knockdown had a negligible effect on accessibility in WT HCT116 cells for any cluster. Clusters 1–3 exhibit increased accessibility upon loss of ARID1A alone or both ARID1A and ARID1B; sites in Cluster 1 gain accessibility in ARID1A$^{-/-}$ ARID1B KD cells, sites in Cluster 2 gain accessibility in ARID1A$^{-/-}$ cells, but lose this accessibility upon ARID1B knockdown, and sites in Cluster 3 gain accessibility in ARID1A$^{-/-}$ cells, which is maintained with ARID1B knockdown. In contrast, Clusters 4 and 6 contain sites that lose accessibility, with unique dependence on ARID1A alone or in combination with ARID1B. Specifically, accessibility at Cluster 4 sites is dramatically reduced in ARID1A$^{-/-}$ HCT116 cells, with minimal further reduction in ARID1A$^{-/-}$ ARID1B KD cells, while accessibility at Cluster 6 sites is slightly reduced in ARID1A$^{-/-}$ cells, and this residual accessibility is completely ARID1B-dependent. Accessible sites in Clusters 5 and 7 are primarily unchanged. Overall, our data demonstrate that ARID1A and ARID1B regulate chromatin accessibility at shared and unique sites across the genome. ARID1A affects a greater proportion

of total accessible regions and maintains open chromatin in ARID1B-deficient cells, whereas ARID1B has a significant function only when ARID1A is lost.

Strikingly, ATAC-seq Clusters positively correlated with significant changes in RNA expression by RNA-seq (*Figure 1F*), highlighting the importance of accessibility in regulating gene transcription. We subjected annotated sites from each cluster group to hierarchical clustering of Hallmark gene sets following Gene Set Enrichment Analysis (*Liberzon et al., 2015*). We found that Clusters 2, 4, and 6 grouped together, consistent with downregulation in ARID1A$^{-/-}$ ARID1B KD cells. These Clusters were enriched for genes involved in KRAS signaling, TNFα signaling via NF-κB, estrogen response, TGFβ signaling, Hedgehog signaling, and epithelial mesenchymal transition (*Figure 1—figure supplement 3*). Notably, HCT116 cells have activating mutations in KRAS and PIK3CA, which share common targets with these gene sets through activation of PI3K/Akt/mTOR, MAPK, and NF-κB pathways that regulate cell cycle progression and proliferation. Clusters 1 and 3 clustered together with prominent enrichment in Hypoxia, while Clusters 5 and 7 were enriched for housekeeping processes, including metabolic and protein homeostasis. These data demonstrate that ARID1A and ARID1B specifically regulate chromatin accessibility and expression of genes that may be crucial for the observed synthetic lethality in HCT116 colorectal cancer cells.

### Requirement for ARID1A or ARID1B is determined by SWI/SNF complex occupancy

To determine whether accessible sites are enriched for SWI/SNF complex occupancy, we overlapped our ATAC-seq data with binding of SMARCA4 or SMARCC1 subunits from ChIP-seq experiments in HCT116 cells (*Mathur et al., 2017*). We found that the majority of SMARCA4 and SMARCC1 binding occurred in accessible regions (*Figure 2A*), consistent with previous reports showing strong overlap between chromatin remodelers and DNA accessibility (*Bao et al., 2015*; *Morris et al., 2014*). Strikingly, SMARCA4 and SMARCC1 binding in WT HCT116 cells was preferentially enriched at decreased sites in Clusters 4 and 6, indicating that SWI/SNF occupancy is a key determinant for sensitivity to ARID1A/1B loss (*Figure 2B*). Previous reports have shown that while the majority of SWI/SNF complex binding sites are lost in ARID1A$^{-/-}$ HCT116, residual ARID1B-containing complexes are intact in ARID1A$^{-/-}$ HCT116 cells (*Figure 1—figure supplement 1D*) and likely contribute to stable and gained SWI/SNF binding in ARID1A$^{-/-}$ HCT116 cells (*Mathur et al., 2017*). To determine if changes in SWI/SNF complex binding are predictive of changes in accessibility, we categorized SMARCA4 and SMARCC1 sites that are lost, gained, or unchanged in ARID1A$^{-/-}$ HCT116 cells according to accessibility cluster (*Figure 2C*). Of the small number of gained SWI/SNF binding sites, the majority are found in Clusters 2 and 3, consistent with a role for SWI/SNF complexes in opening chromatin. Sites that lose SWI/SNF binding fall within Clusters 4 and 6, while stable sites are enriched in Clusters 4, 6 and 7 (*Figure 2C*). Thus, changes in SWI/SNF complex binding correlate with changes in accessibility in WT versus ARID1A$^{-/-}$ HCT116 cells. Furthermore, the residual binding of SWI/SNF complexes was predictive of ARID1B dependence in ARID1A$^{-/-}$ HCT116 cells. For example, while SWI/SNF binding is dramatically reduced at Cluster 4 sites in ARID1A$^{-/-}$ HCT116 cells, there is residual binding of SMARCA4 and SMARCC1 at Cluster 6 sites, consistent with a requirement for both ARID1A and ARID1B (*Figure 2B*). Genome-wide, we find that accessibility at SMARCA4 or SMARCC1-bound accessible sites is highly sensitive to loss of both proteins (*Figure 2D*). These data indicate that dependency on ARID1A or ARID1B is determined by SWI/SNF complex binding in WT and ARID1A$^{-/-}$ HCT116 cells, respectively. Increased accessible sites gain SWI/SNF complex binding in ARID1A$^{-/-}$ HCT116 cells, while decreased accessible sites have the highest enrichment of SWI/SNF complex binding in both settings. This characteristic may engender these sites as particularly vulnerable to loss of ARID1A or ARID1B.

### ARID1A/1B-dependent accessible sites are primarily located in enhancers

To gain a clearer picture of how ARID1A- and ARID1B-dependent accessible sites are distributed across the genome, we examined the overlap of Clusters 1–7 with ChIP-seq peak calling of histone modifications (*Figure 3A*). Strikingly, we observed that decreased accessible sites in Clusters 4 and 6 are highly enriched for H3K4me and H3K27ac enhancer marks, but depleted for promoter-specific H3K4me3, suggesting that loss of ARID1A/1B primarily affects accessibility at enhancers. In contrast,

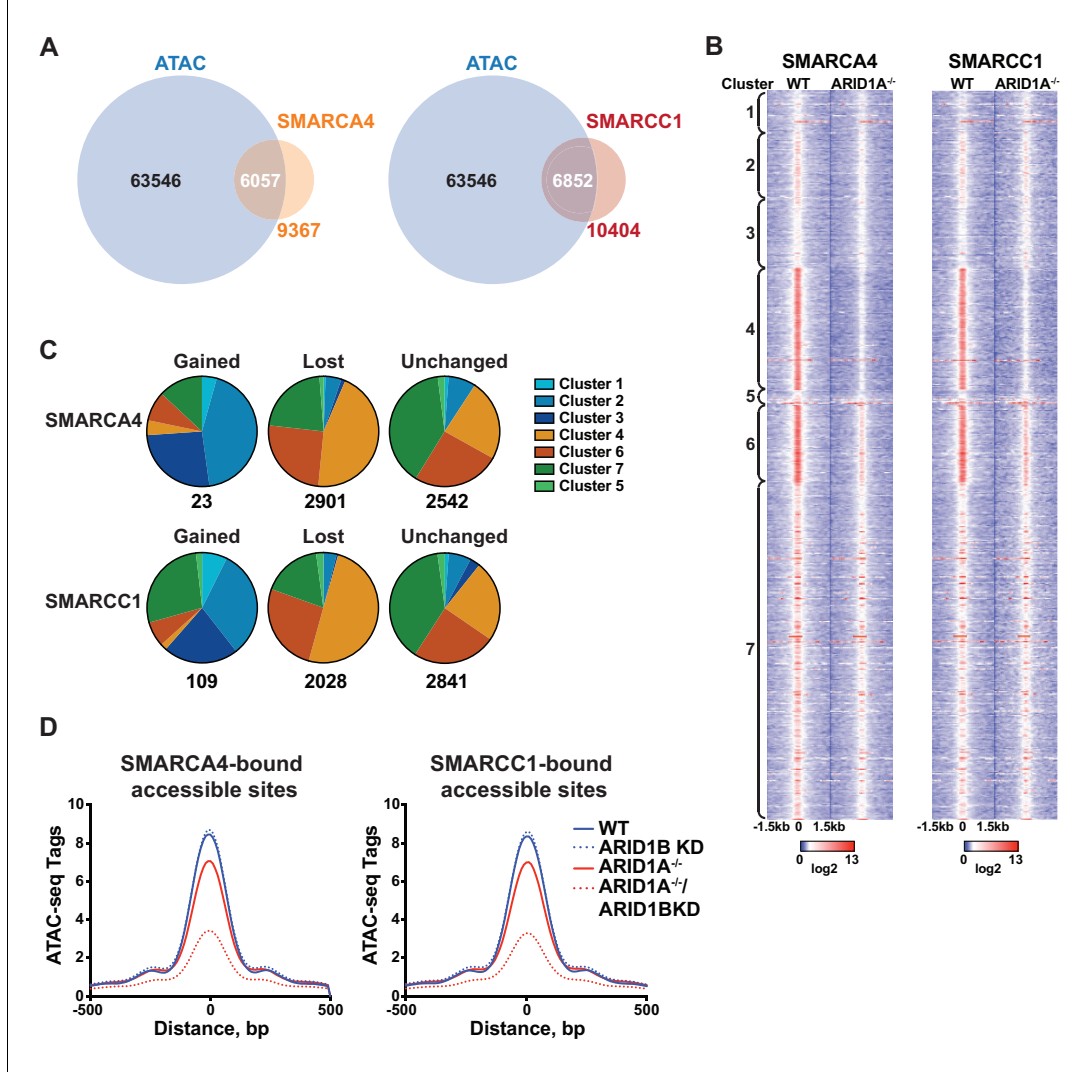

**Figure 2.** SWI/SNF occupancy correlates with ARID1A/1B-dependent changes in accessibility. (**A**) Venn diagram overlap of SMARCA4 or SMARCC1 ChIP binding sites with accessible sites in WT HCT116 cells. (**B**) ChIP-seq tag density of SMARCA4 or SMARCC1 from WT and ARID1A-/- HCT116 cells centered on accessible sites in Clusters 1–7 ± 1.5 kb. (**C**) Distribution of SMARCA4 or SMARCC1 binding sites that are gained, lost, or unchanged in ARID1A$^{-/-}$ versus WT HCT116 cells across ATAC-seq Clusters 1–7. Number of total gained, lost, or unchanged SMARCA4/SMARCC1-bound accessible sites is shown. (**D**) ATAC-seq tag density from HCT116 WT, ARID1B KD, ARID1A$^{-/-}$, and ARID1A$^{-/-}$/ARID1B KD cells at SMARCA4 or SMARCC1-bound accessible sites.

DOI: https://doi.org/10.7554/eLife.30506.008

increased accessible sites in Clusters 1 and 3 and unchanged sites in Clusters 5 and 7 were enriched for H3K4me3 found at promoters (*Figure 3A and B*). None of the clusters is enriched for H3K27me3, consistent with little overlap of this repressive mark with accessible chromatin. We then examined whether the requirement for ARID1A/1B-dependent remodeling is associated with enhancer activity. Specifically, H3K27ac has been used to distinguish poised enhancers (H3K27ac-) from active enhancers (H3K27ac+) and super enhancers (H3K27ac high) (*Rada-Iglesias et al., 2011*; *Whyte et al., 2013*). Based on these criteria, we find that sites with decreased accessibility in ARID1A$^{-/-}$ ARID1B KD HCT116 cells are enriched at all three enhancer classes, but primarily at poised and active enhancers, while increased accessible sites are enriched at promoters and 5' untranslated regions (*Figure 3B and C*). We then profiled nascent transcription at distal sites or promoters against the change in accessibility in WT versus ARID1A$^{-/-}$ ARID1B KD HCT116 cells by referencing published GRO-seq data in HCT116 cells (GSE38140, *Galbraith et al., 2013*)

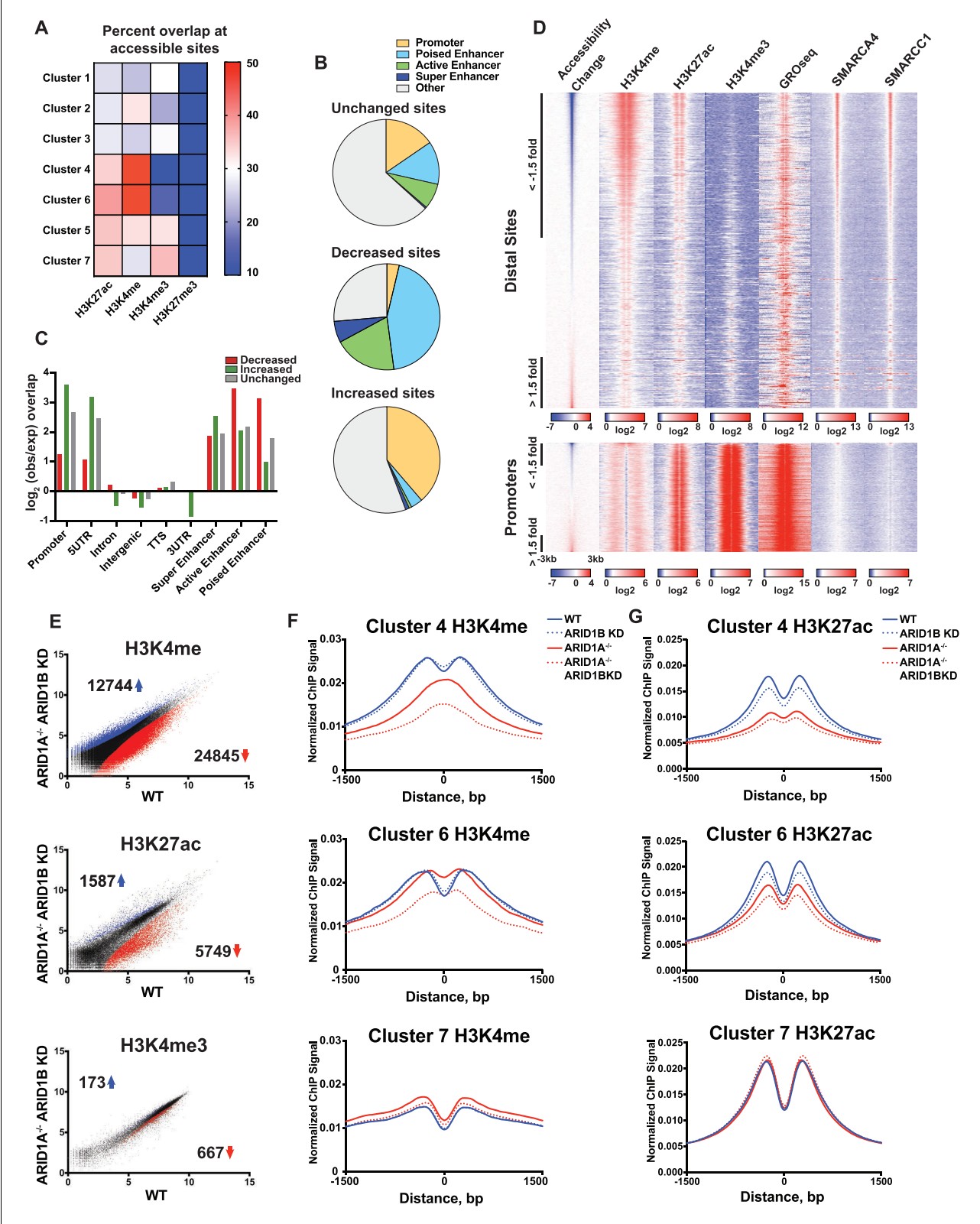

**Figure 3.** ARID1A/1B maintain accessibility at enhancers. (A) Proportion of accessible sites in Clusters 1–7 that directly overlap with ChIP-seq regions for histone 3 lysine 27 acetylation (H3K27ac), histone 3 lysine 4 monomethylation (H3K4me), histone 3 lysine 4 trimethylation (H3K4me3), and histone 3 lysine 27 trimethylation (H3K27me3) in WT HCT116 cells. Overlaps ≥0.5 were marked in red; overlaps ≤0.1 were marked in blue. Intergenic GRO-seq signal at Clusters 1–7 reveals bi-directional transcription at accessible sites (*Figure 3—figure supplement 1*). (B) Percent of unchanged, increased, and

*Figure 3 continued on next page*

*Figure 3 continued*

decreased accessible sites called from WT versus ARID1A$^{-/-}$ ARID1B KD HCT116 cells present in promoters, poised enhancers, active enhancers, super enhancers, or other genomic elements. (C) Observed/expected enrichment of unchanged, increased, and decreased accessible sites called from WT versus ARID1A$^{-/-}$ ARID1B KD HCT116 cells at promoters, 5' and 3' untranslated regions (UTR), introns, intergenic regions, transcription termination sites (TTS), poised enhancers, active enhancers, and super enhancers. (D) Accessible sites sorted by log2 fold change in accessibility in WT versus ARID1A$^{-/-}$ ARID1B KD HCT116 cells at distal sites and promoters, where distal is defined as greater than 3 kb from the nearest TSS. Corresponding H3K4me, H3K27ac, H3K4me3, GRO-seq, SMARCA4, and SMARCC1 tag density from WT HCT116 cells is displayed. Reads are centered on middle of accessible region ±3 kb. (E) Differential peak calls from H3K27ac, H3K4me, or H3K4me3 ChIP-seq from HCT116 WT cells expressing shRNAs to scrambled control (WT) versus HCT116 ARID1A$^{-/-}$ cells expressing shRNAs to ARID1B (ARID1A$^{-/-}$ ARID1B KD). Blue and red dots represent differential ChIP-seq peaks whose read density increased or decreased by 1.5 fold or more (FDR < 0.05), respectively, from two independent biological replicates. Numbers in the upper left and lower right corners refer to numbers of increased and decreased peaks. Breakdown of H3K4me3 changes in Clusters 1–7 can be found in *Figure 3—figure supplement 2A,B*. (F) Normalized average ChIP-seq tag density for H3K4me from two independent biological replicates at accessible sites in Clusters 4, 6, and 7 from HCT116 WT, ARID1A$^{-/-}$, ARID1B KD, and ARID1A$^{-/-}$ ARID1B KD cells. Reads are centered on middle of accessible region ±1.5 kb. Data for all 7 clusters can be found in *Figure 3 – Figure Supplement 3A, B*. (G) Normalized average ChIP-seq tag density for H3K27ac from two independent biological replicates at accessible sites in Clusters 4, 6, and 7 from HCT116 WT, ARID1A$^{-/-}$, ARID1B KD, and ARID1A$^{-/-}$ ARID1B KD cells. Reads are centered on middle of accessible region ±1.5 kb. Data for all 7 clusters can be found in *Figure 3—figure supplement 3C,D*.

DOI: https://doi.org/10.7554/eLife.30506.009

The following figure supplements are available for figure 3:

**Figure supplement 1.** GRO-seq signal from intergenic sites in Clusters 1–7 centered on the middle of the accessible region ±2 kb.
DOI: https://doi.org/10.7554/eLife.30506.010

**Figure supplement 2.** Analysis of H3K4me3 ChIP-seq reads at accessible sites in Clusters 1-7,
DOI: https://doi.org/10.7554/eLife.30506.011

**Figure supplement 3.** Analysis of H3K4me and H3K27ac ChIP-seq reads at accessible sites in Clusters 1-7,
DOI: https://doi.org/10.7554/eLife.30506.012

(*Figure 3D*). Nascent transcription giving rise to enhancer RNAs (eRNAs) has been observed at active enhancers and is thought to facilitate gene transcription through direct and indirect mechanisms (*Arner et al., 2015*; *Hah et al., 2013*; *Maruyama et al., 2014*; *Mousavi et al., 2013*). As observed, ARID1A/1B-dependent decreased accessible sites exhibit high levels of SMARCA4 and SMARCC1 binding, as well as strong enrichment for H3K4me and H3K27ac marks (*Figure 3D*). In addition, downregulated sites are enriched for nascent transcription, indicating that ARID1A/1B loss impacts accessibility at actively transcribing enhancers. After parsing intergenic sites in Clusters 1–7, we found that Cluster 6 intergenic regions have higher levels of nascent transcription than Cluster 4 sites, consistent with the notion that Cluster 4 sites are less active than Cluster 6 sites (*Figure 3—figure supplement 1*). Sensitivity at these sites may be due to their stronger dependence on the SWI/SNF remodeling function.

## ARID1A/1B are necessary to maintain active enhancer architecture

We next asked whether loss of ARID1A and ARID1B directly affects properties of enhancer or promoter architecture. To this end, we performed ChIP-seq for H3K27ac, H3K4me, and H3K4me3 in WT, ARID1B KD, ARID1A$^{-/-}$, and ARID1A$^{-/-}$ ARID1B KD HCT116 cells. Overall, there were thousands of differential H3K27ac and H3K4me sites that increased or decreased between WT and ARID1A$^{-/-}$ ARID1B KD HCT116 cells, while only a handful of sites showed differential H3K4me3 marking (*Figure 3E*, *Figure 3—figure supplement 2*). We found that ARID1A and ARID1B-dependent accessible sites in Clusters 4 and 6 had reduced H3K4me and H3K27ac marks in ARID1A$^{-/-}$ ARID1B KD HCT116 cells, while unaltered accessible sites in Cluster 7 did not change (*Figure 3F and G*). Consistent with the strong dependence of Cluster 4 accessible sites on ARID1A, H3K4me and H3K27ac were markedly reduced in ARID1A$^{-/-}$ cells, and further reduced with ARID1B loss. In contrast, ARID1A removal repressed H3K27ac but not H3K4me at Cluster 6 accessible sites, while knockdown of ARID1B in ARID1A$^{-/-}$ cells caused loss of both marks (*Figure 3F and G*). Small changes in H3K4me are evident at sites in Clusters 1–3, which become more accessible in ARID1A$^{-/-}$ or ARID1A$^{-/-}$ ARID1B KD HCT116 cells (*Figure 3—figure supplement 3*). These data support a role for ARID1A and ARID1B in maintaining accessibility and active histone marks at enhancer regions. Although the SWI/SNF complex has previously been implicated in maintaining H3K27ac (*Bao et al., 2015*; *Mathur et al., 2017*; *Raab et al., 2015*), a function for ARID1B in accessibility and H3K27ac marks at

enhancer regions is unprecedented. Similarly, clustering accessible sites by strength of ARID1A and ARID1B regulation has revealed novel roles for SWI/SNF in maintenance of H3K4me, consistent with the requirement for SWI/SNF complex remodeling at both poised and active enhancers.

## ARID1A/1B are required for binding of AP-1 complex family members

SWI/SNF complexes are generally recruited by transcription factors and facilitate further transcription factor binding, altering nucleosome spacing to increase chromatin accessibility around these sites. We therefore sought to determine whether occupancy of transcription factors is disrupted at ARID1A- and ARID1B-dependent sites. We performed hierarchical clustering of enriched motifs from clustered accessible sites (*Figure 4—figure supplement 1*). We identified motifs for AP-1 members FRA1, FOSL2, and ATF3 as significantly enriched in both Cluster 4 and 6 (*Figure 4A and B*), while CTCF motifs were enriched in all Clusters (*Figure 4—figure supplement 1*). Using ChIP-seq data for FRA1, JUND, CTCF, and ATF3 available from ENCODE, we found that FRA1, JUND, ATF3, and CTCF were bound at 41%, 39%, 16%, and 19% of decreased accessible sites, respectively (*Figure 4C*). Indeed, FRA1 and JUND had the strongest binding at Cluster 4 and 6 sites and CTCF binding was present throughout, but depleted in Cluster 4 (*Figure 4D*). Accessible sites found at FRA1 or JUND binding sites from Cluster 4 and 6 showed reduced accessibility in ARID1A$^{-/-}$ cells and ARID1A$^{-/-}$ ARID1B KD cells, consistent with overall dependence of Cluster 4 and 6 sites on ARID1A alone or in combination with ARID1B (*Figure 4E and F*). We next investigated accessibility around AP-1 motifs to determine if loss of SWI/SNF binding influences nucleosome architecture. We measured nucleosome spacing around the AP-1 motif at Cluster 4 and 6 sites by selecting mononucleosome-sized ATAC-seq reads from each genotype. This analysis revealed a stepwise reduction in distance between nucleosome peaks with ARID1A loss and further ARID1B removal (*Figure 4G*). A total 37–40 base pair reduction in the distance between mononucleosome peaks was observed at ARID1A- and ARID1B-dependent accessible sites, suggesting a reduction of nucleosome spacing around the AP-1 motif. Consistent with diminished opening, FRA1 occupancy was severely reduced at Cluster 4 and 6 sites in ARID1A$^{-/-}$ and ARID1A$^{-/-}$ ARID1B KD HCT116 cells at *TGFA*, *NRIP1*, *PLAU*, and *CALB2*, which are downregulated in ARID1A$^{-/-}$ ARID1B KD cells (*Figure 4H*). These data suggest that the SWI/SNF complex controls occupancy of the AP-1 transcription factor, potentially by influencing chromatin accessibility, and implies cooperation of these two complexes in regulating expression of nearby target genes.

## ARID1A and ARID1B regulate gene expression by maintaining active enhancer architecture

To understand how ARID1A/1B-dependent changes influence gene expression, we examined differentially expressed genes between WT and ARID1A$^{-/-}$ ARID1B KD cells (*Figure 5A*). Loss of ARID1A and ARID1B significantly reduced expression of 1947 genes and enhanced expression of 1311 genes. 48% of the differentially called down genes had at least one decreased accessible site that mapped to the TSS in the ARID1A$^{-/-}$ ARID1B KD cells, while 23% of the upregulated genes had at least one increased accessible site. We next examined how many genes are affected by ARID1A/1B-dependent changes in accessibility and H3K27ac at sites directly bound by SWI/SNF and FRA1, focusing on downregulated accessible sites due to their stronger concordance with gene expression (*Figure 1F*). Specifically, 4909 (39%) of differentially decreased ATAC sites overlapped with SWI/SNF binding (*Figure 5B*). Of these, 2156 sites also exhibited reduced H3K27ac marks and a further 1709 were also bound by FRA1. We annotated these co-regulated sites by mapping to the nearest TSS, and found that 377 of these overlapped with genes downregulated in ARID1A$^{-/-}$ ARID1B KD cells. This number is likely an underestimate as it does not include genes that did not meet our 1.5 fold, FDR < 0.05 threshold. This subset includes active enhancer regulatory regions engaged in eRNA transcription, as demonstrated by higher GRO-seq signal at co-regulated sites (ATAC Down, H3K27ac Down, SWI/SNF Bound, FRA1 Bound) than at decreased accessible sites that lack SWI/SNF, H3K27ac, and FRA1 binding (*Figure 5C*). We next used RNA Pol II ChIA-PET data from the ENCODE project to determine whether co-regulated sites directly interact with gene promoters and other regulatory regions to affect gene expression. We compared how often co-regulated sites are coupled $^{to}$ a change in accessibility, H3K27 acetylation, or transcription at an interacting site (*Figure 5D*). Co-regulated sites are enriched for interactions with sites that lose accessibility or

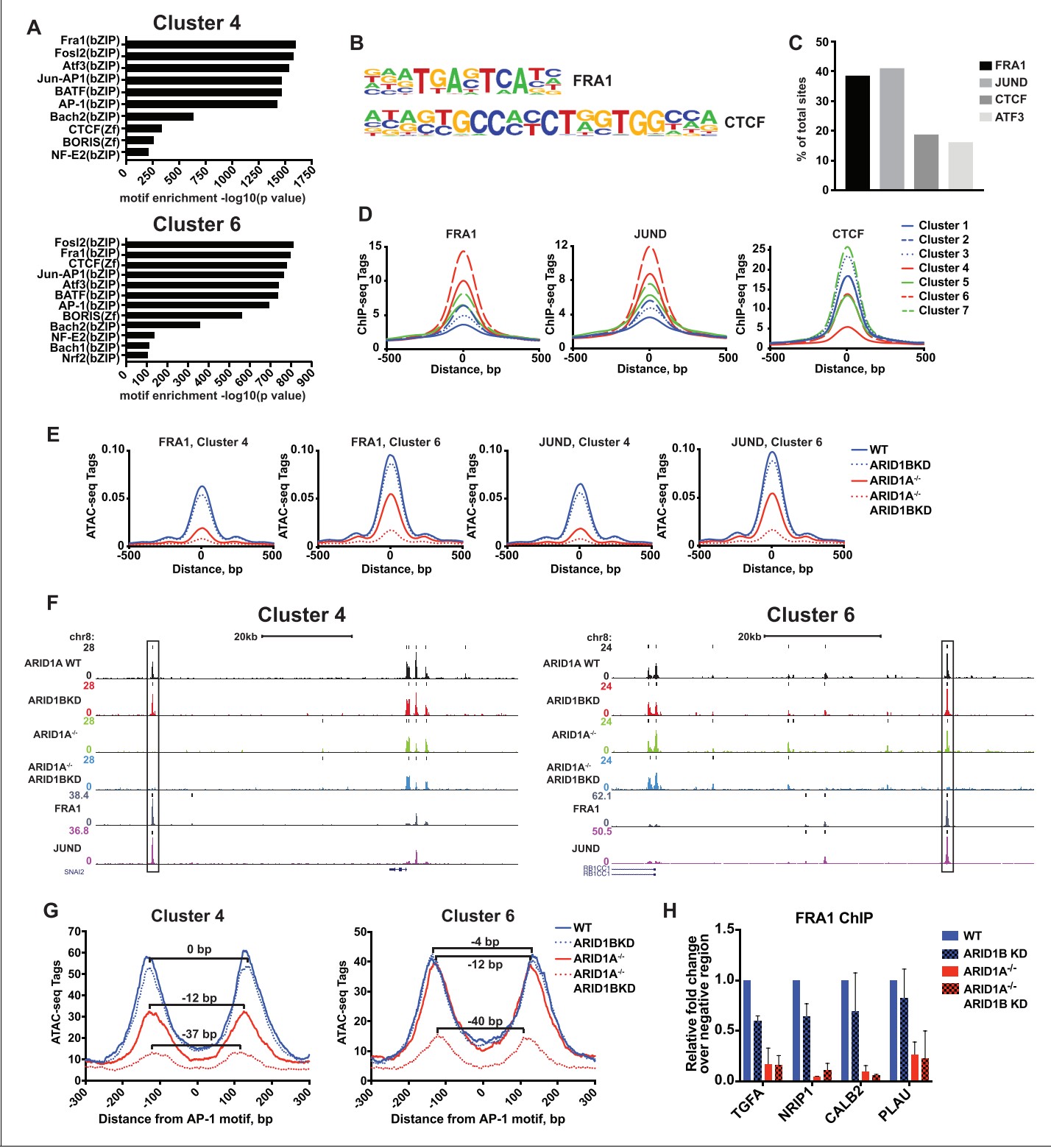

**Figure 4.** ARID1A/1B are required for the binding of AP-1 transcription factors at accessible sites. (**A**) Significance of motif enrichment of known motifs on Cluster 4 and 6 accessible sites. Motif enrichment for Clusters 1–7 can be found in *Figure 4—figure supplement 1*. (**B**) Known FRA1 and CTCF motif from HOMER database. (**C**) Proportion of decreased accessible sites in WT versus ARID1A$^{-/-}$ ARID1B KD HCT116 cells with overlapping ChIP-seq peaks for FRA1, JUND, CTCF and ATF3 from ENCODE ChIP-seq datasets. (**D**) FRA1, JUND, and CTCF ChIP-seq occupancy at accessible sites in Clusters 1–7. (**E**) ATAC-seq tag density at FRA1 and JUND binding sites in Clusters 4 and 6 from WT, ARID1B KD, ARID1A$^{-/-}$, and ARID1A$^{-/-}$ ARID1B KD

*Figure 4 continued on next page*

*Figure 4 continued*

HCT116 cells. (F) Genome browser tracks of ATAC-seq data (top four tracks) from WT, ARID1B KD, ARID1A$^{-/-}$, and ARID1A$^{-/-}$ ARID1B KD HCT116 cells and ChIP-seq data from WT HCT116 ENCODE data (bottom two tracks). Representative Cluster 4 and Cluster 6 sites overlapping FRA1 and JUND binding sites are indicated with a black box. (G) Nucleosome spacing for mononucleosome fragments at AP-1 motifs in Clusters 4 and 6. Distance between mononucleosome peaks in ARID1B KD, ARID1A$^{-/-}$, and ARID1A$^{-/-}$ ARID1B KD HCT116 cells was calculated relative to WT HCT116 cells. (H) FRA1 binding by ChIP-qPCR at two Cluster 4 sites (*TGFA*, *NRIP1*) and two Cluster 6 sites (*CALB2*, *PLAU*) in ARID1B KD, ARID1A$^{-/-}$, and ARID1A$^{-/-}$ ARID1B KD HCT116 cells relative to WT HCT116 cells. Values shown are averages from two biological replicates where error bars indicate standard deviation.

DOI: https://doi.org/10.7554/eLife.30506.013

The following source data and figure supplement are available for figure 4:

**Source data 1.** FRA1 occupancy by ChIP-qPCR in WT and ARID1A$^{-/-}$ HCT116 cells with ARID1B knockdown.

DOI: https://doi.org/10.7554/eLife.30506.015

**Figure supplement 1.** Heatmap of motif enrichment p values (log$_e$) in Clusters 1–7.

DOI: https://doi.org/10.7554/eLife.30506.014

H3K27ac, but anti-correlated with sites that gain these features. In addition, co-regulated sites are more likely to interact with TSSs of genes that lose expression, as shown for *TGFA*, *TGFBR2*, *PLAU*, and *EGFR* (*Figure 5—figure supplement 1*), but not gain expression. The inverse is true for increased accessible sites, which are more likely to interact with sites with increased accessibility, increased H3K27ac, and increased expression (*Figure 5D*, *Figure 5—figure supplement 2*). These data indicate that ARID1A/1B-dependent changes in accessibility and H3K27ac at enhancers strongly impact gene expression as a result of looping interactions with TSSs and other regulatory regions.

## ARID1A/1B maintain transcription of genes in cancer growth promoting pathways, including *MET*

We then profiled up and downregulated genes from all four genotypes (*Figure 5E*), focusing on significant changes between ARID1A$^{-/-}$ and ARID1A$^{-/-}$ ARID1B KD cells to uncover factors that could contribute to the synthetic lethal phenotype observed between these SWI/SNF subunits. Hierarchical clustering grouped ARID1B KD with the WT expression profile as expected, while ARID1A$^{-/-}$ and ARID1A$^{-/-}$ ARID1B KD expression profiles clustered together, suggesting that ARID1B KD largely enhances the effect of ARID1A loss. However, an additional 16 genes were differentially upregulated and 96 genes downregulated between ARID1A$^{-/-}$ and ARID1A$^{-/-}$ ARID1B KD cells. KEGG analysis of ARID1A/1B targets revealed enrichment in Pathways in Cancer, with specific identification of Adherens junction, Ras/Rap1 signaling pathway, ECM-receptor interaction, PI3K-AKT signaling pathway, and ErbB signaling pathway (*Figure 5—figure supplement 3*). In particular, growth response pathways through receptor tyrosine kinases (RTKs) were significantly downregulated in ARID1A$^{-/-}$ ARID1B KD cells, including genes encoding ligands and receptors in the EGFR pathway and related MET, IGFR, VEGFR, PDGFR, FGFR, AXL, and IL6R pathways as well as signaling intermediates of the downstream KRAS and PI3K pathways, like *PIK3R1* (*Figure 5E*). Reduced KRAS/PI3K-dependent signaling corresponded with downregulation of activated transcription factors including *JUN*, *JUND*, *FOSB*, *MYC*, and *ETS1/2*, and reduced expression of proliferation genes *CCND1* and *CDK6* and anti-apoptotic factors *BCL2* and *BCL2L1*.

We treated these genes as potential key contributors to the observed synthetic lethal phenotype in ARID1A$^{-/-}$ ARID1B KD cells. Specifically, we surveyed the genes most strongly reduced by ARID1B KD in ARID1A$^{-/-}$ cells and identified *MET,* a known driver of proliferation, as a potentially crucial target of ARID1A and ARID1B in colorectal cancer. *MET* codes for the MET receptor tyrosine kinase, which is activated by hepatocyte growth factor (HGF) and recruits downstream signaling intermediates including PI3K, SRC and STAT3 (*Boccaccio et al., 1998*; *Ponzetto et al., 1994*; *Zhang et al., 2002*). MET is highly expressed in ~67% of primary colorectal carcinomas (*Gayyed et al., 2015*) and overexpression is associated with tumor progression and metastasis (*Baldus et al., 2007*; *Luo and Xu, 2014*; *Zeng et al., 2004*). MET is necessary for proliferation of colorectal carcinomas (*Li et al., 2014*), and selective inhibition of MET with small molecule inhibitors is effective in reducing proliferation of HCT116 cells (*Larsen and Dashwood, 2010*). In HCT116 cells, we identified multiple accessible regions from Clusters 4 and 6 that are co-bound by the AP-1 complex and the SWI/SNF

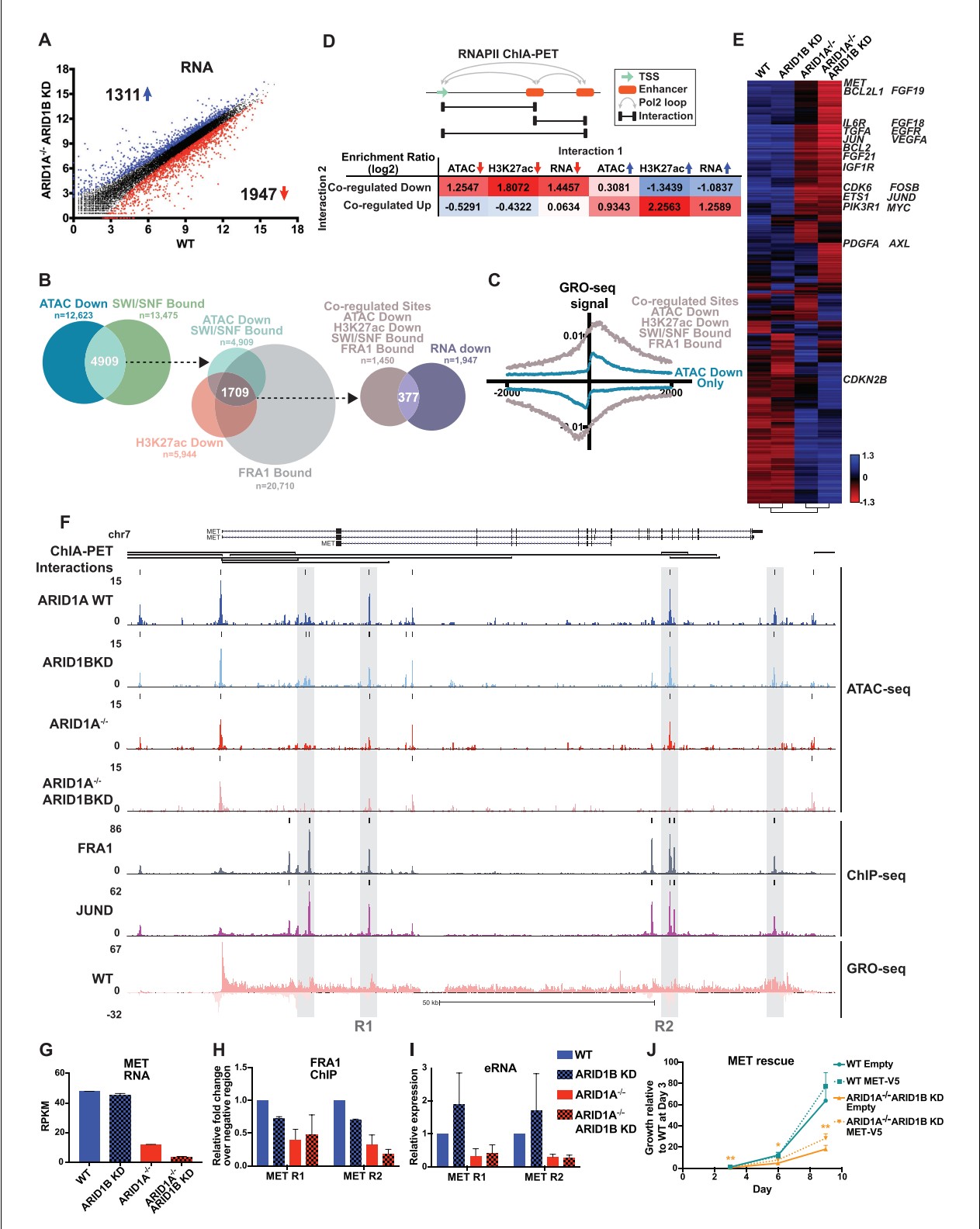

**Figure 5.** ARID1A/1B regulate gene expression of cell growth pathway genes by maintaining active enhancer architecture. (**A**) Differentially expressed genes in WT versus ARID1A⁻/⁻ ARID1B KD HCT116 cells. Blue and red dots represent differential RNA-seq expression increased or decreased by 1.5 fold or more (FDR < 0.05), respectively, from two independent biological replicates. Numbers in upper left and lower right hand corners indicate number of differentially expressed genes. (**B**) Genomic locations with reduced accessibility in ARID1A⁻/⁻ ARID1B KD HCT116 cells (ATAC Down) were

*Figure 5 continued on next page*

*Figure 5 continued*

overlapped with SMARCA4 and/or SMARCC1 binding sites identified in WT HCT116 cells (SWI/SNF Bound). Common positions (ATAC Down, SWI/SNF Bound) were then overlapped with reduced H3K27ac sites in ARID1A$^{-/-}$ ARID1B KD cells (H3K27ac Down) and FRA1-bound sites identified in WT HCT116 cells (FRA1 Bound). Co-regulated sites (ATAC Down, H3K27ac Down, SWI/SNF Bound, FRA1 Bound) were annotated and overlapped with genes downregulated in ARID1A$^{-/-}$/ARID1B KD cells (RNA Down). 377 genes were identified with SWI/SNF- and FRA1 binding and decreased accessibility, H3K27ac and RNA expression. Examples of co-regulated down sites in and around *TGFA, TGFBR2, PLAU,* and *EGFR* can be found in ***Figure 5—figure supplement 1***. (C) GRO-seq signal at co-regulated sites (ATAC Down, H3K27ac Down, SWI/SNF Bound, FRA1 Bound) compared to sites that are not bound by SWI/SNF or FRA1 but demonstrated reduced accessibility (ATAC Down Only). (D) Observed/expected enrichment ratio (log2) for interactions between transcription start sites (TSSs) and regulatory regions through chromatin looping. Sites bound by SWI/SNF and FRA1 and exhibiting reduced accessibility and H3K27ac (Co-regulated Down) were compared to sites with decreased/increased accessibility (ATAC Down/Up), sites with decreased/increased H3K27ac (H3K27ac Down/Up), or TSSs of genes with decreased/increased expression (RNA Down/Up). Examples of interacting co-regulated down sites in and around *TGFA, TGFBR2, PLAU, and EGFR* can be found in ***Figure 5—figure supplement 1***. The same analysis was applied to sites exhibiting increased accessibility and H3K27ac (Co-regulated Up). Examples of interacting co-regulated up sites in and around *ANXA10, BMP4, WLS, and OBFC1* can be found in ***Figure 5—figure supplement 2***. (E) Hierarchical clustering of 7890 genes increased or decreased by ≥1.5 fold gene expression from RNA-seq of WT, ARID1B KD, ARID1A$^{-/-}$, and ARID1A$^{-/-}$ARID1B KD HCT116 cells. KEGG pathway analysis of affected genes can be found in ***Figure 5—figure supplement 3***. (F) Regulatory regions in and around the *MET* gene. Gray boxes indicate sites with decreased accessibility in ARID1A$^{-/-}$ ARID1B KD HCT116 cells that are bound by the AP-1 complex and have GRO-seq signal. Not shown: ChIP-seq tracks demonstrating that gray boxed sites are also bound by SMARCA4 and SMARCC1 and have decreased H3K27ac in ARID1A$^{-/-}$ ARID1B KD HCT116 cells. (G) *MET* expression from RNA-seq. Error bars indicate standard deviation for two independent replicates. (H) FRA1 ChIP levels determined by ChIP-qPCR at R1 and R2 (see 5F) within *MET*. Averages from two independent experiments are shown with standard deviation. (I) Enhancer RNA levels determined by RT-qPCR at R1 and R2 (see 5F) within *MET*. Averages from three independent experiments are shown with standard deviation. (J) Growth of WT and ARID1A$^{-/-}$ ARID1B KD cells with forced expression of MET (MET-V5) or empty vector (Empty) as control. Protein expression of MET and V5 is shown in ***Figure 5—figure supplement 4***. Averages from three replicates from two different infections. *p<0.05; **p<0.01.
DOI: https://doi.org/10.7554/eLife.30506.016

The following source data and figure supplements are available for figure 5:

**Source data 1.** FRA1 occupancy by ChIP-qPCR at *MET* regions in WT and ARID1A$^{-/-}$ HCT116 cells with ARID1B knockdown.
DOI: https://doi.org/10.7554/eLife.30506.021
**Source data 2.** eRNA level at *MET* regions in WT and ARID1A$^{-/-}$ HCT116 cells with ARID1B knockdown.
DOI: https://doi.org/10.7554/eLife.30506.022
**Source data 3.** Proliferation of cells with forced *MET* expression in WT and ARID1A$^{-/-}$ HCT116 cells with ARID1B knockdown.
DOI: https://doi.org/10.7554/eLife.30506.023
**Source data 4.** MET protein expression in WT and ARID1A$^{-/-}$ HCT116 cells with ARID1B knockdown.
DOI: https://doi.org/10.7554/eLife.30506.024
**Source data 5.** MET protein expression in WT and ARID1A$^{-/-}$ HCT116 cells with ARID1B knockdown with forced *MET* expression.
DOI: https://doi.org/10.7554/eLife.30506.025
**Figure supplement 1.** Genome browser tracks of representative downregulated genes in ARID1A$^{-/-}$ ARID1B KD cells.
DOI: https://doi.org/10.7554/eLife.30506.017
**Figure supplement 2.** Genome browser tracks of representative upregulated genes in ARID1A$^{-/-}$ ARID1B KD cells.
DOI: https://doi.org/10.7554/eLife.30506.018
**Figure supplement 3.** KEGG pathway analysis of ARID1A/1B-dependent genes reveals enrichment in Ras/PI3K-Akt signaling pathways.
DOI: https://doi.org/10.7554/eLife.30506.019
**Figure supplement 4.** MET protein expression is ARID1A/1B-dependent in HCT116 cells, but can be reconstituted by forced expression of MET.
DOI: https://doi.org/10.7554/eLife.30506.020

complex within and downstream of the *MET* gene (***Figure 5F***, data not shown). These areas exhibited bi-directional GRO-seq signal and looping interactions back to the promoter and other regulatory sites. We focused on two regions (R1 and R2) with reduced accessibility in ARID1A$^{-/-}$ cells that was completely abolished by ARID1B loss, consistent with greater than 70% reduction in RNA expression (***Figure 5G***) and total protein level (***Figure 5—figure supplement 4A,B***). A previous study has shown that the *MET* gene is transactivated by the AP-1 complex (***Seol et al., 2000***), and here loss of ARID1A and ARID1B proteins inhibited binding of FRA1 at two regulatory regions (***Figure 5H***), similar to the effects we observed at four other ARID1A/1B-dependent genes (***Figure 4H***). Enhancer RNA expression was also strongly reduced in ARID1A$^{-/-}$ and ARID1A$^{-/-}$ ARID1B KD cells (***Figure 5I***). Forced expression of MET (***Figure 5—figure supplement 4C,D***) led to a significant increase in the proliferation of ARID1A$^{-/-}$ ARID1B KD cells (***Figure 5J***), suggesting that loss of MET expression is a contributing factor to the synthetic lethal phenotype. However, rescue of MET alone was insufficient to completely counteract the proliferation defect caused by ARID1A and

ARID1B loss, which is likely due to the downregulation of multiple factors. Taken together, these data suggest that ARID1A and ARID1B maintain proliferation of colorectal cancer cells at least in part by sustaining MET oncogenic signaling, specifically by controlling accessibility, transcription factor occupancy, and eRNA transcription at multiple regulatory regions within the *MET* gene.

In all, these results demonstrate a role for SWI/SNF complexes in maintaining accessibility for transcription factor binding and the deposition of active histone marks at enhancers (*Figure 6*). The absence of SWI/SNF complexes dysregulates these processes, resulting in altered expression of key genes in cancer pathways, consistent with reduced cancer cell growth observed upon loss of ARID1A and ARID1B.

## ARID1B regulates accessibility and active histone modifications at enhancers in ARID1A mutant ovarian cancer cells

A synthetic lethal relationship between ARID1A and ARID1B has previously been observed for proliferation of ovarian clear cell carcinoma cells (OCCCs) (*Helming et al., 2014*). We therefore asked whether the mechanism of ARID1A and ARID1B synthetic lethality in HCT116 colorectal carcinoma cells identified in our study is conserved in OCCCs with naturally occurring ARID1A mutations. To this end, we performed ATAC-seq in the ARID1A-mutant TOV21G cell line infected with shRNAs to ARID1B (ARID1B KD) or with a scrambled control (scr). Knockdown of ARID1B led to a 70–80% reduction in ARID1B protein, which impaired cancer cell growth as previously reported (*Figure 7— figure supplement 1A and B*) (*Helming et al., 2014*). By ATAC-seq, we identified 1896 sites with increased accessibility and 1028 sites with decreased accessibility by 1.5 fold or greater upon ARID1B knockdown (FDR < 0.05) (*Figure 7A*). Changes in accessibility were predictive of significant alterations in gene expression for both up- and down-regulated sites (*Figure 7B*). To better understand the mechanism of gene regulation by ARID1B in this context, we further characterized accessible sites altered by ARID1B KD. Consistent with our results in the HCT116 cell line, we found that decreased sites are enriched at active and poised enhancers, but depleted at promoters, while increased sites are enriched at promoters and active enhancers (*Figure 7C and D*). Motifs for AP-1 family members FRA1, ATF3, BATF, and FOSL2 (*Figure 7E*) were significantly enriched at decreased accessible sites, similar to decreased sites in HCT116 cells. Accordingly, accessibility at AP-1 motif sites was also decreased in TOV21G cells following ARID1B knockdown (*Figure 7F*).

To determine how ARID1B knockdown influences histone modifications in OCCCs, we performed ChIP-seq analysis of H3K27ac, H3K4me, and H3K4me3 in scrambled control or ARID1B KD TOV21G cells. We identified thousands of sites with enhanced or repressed modification levels upon ARID1B loss (*Figure 7G*). Decreased accessible sites have high levels of enhancer-associated H3K4me and H3K27ac marks, which are lost upon ARID1B knockdown; in contrast, H3K4me3 levels are initially low and do not change (*Figure 7H*). Increased accessible sites, on the other hand, are high for H3K4me3 and H3K27ac marks that are found at promoters and active enhancers, but the levels of

| Cluster | Total sites | Basal accessibility | Accessibility trend | | H3K27ac trend | H3K4me trend | GROseq signal | SWI/SNF level | FRA1 level |
|---|---|---|---|---|---|---|---|---|---|
| | | | 1A KO | 1B KD | 1A KO + 1B KD | 1A KO + 1B KD | | | |
| 1 | 3,016 | | − | ↑ | ↑ | ↑ | + | − | − |
| 2 | 4,943 | | ↑ | ↓ | − | ↑/↓ | + | + | + |
| 3 | 4,772 | | ↑ | − | ↑ | ↑ | + | − | + |
| 4 | 8,581 | | ↓↓ | ↓ | ↓↓ | ↓↓ | + | +++ | ++ |
| 5 | 1,437 | | ↓ | − | − | − | ++ | + | + |
| 6 | 5,287 | | ↓ | ↓↓ | ↓ | ↓ | ++ | +++ | +++ |
| 7 | 24,532 | | − | − | − | − | +++ | ++ | ++ |

**Figure 6.** Summary of ARID1A/1B dependent changes in Clusters 1–7. Blue arrows indicate increased, red arrows indicate decreased. Signal strength is represented as negative (-—), low (+), moderate (++), and high (+++).
DOI: https://doi.org/10.7554/eLife.30506.026

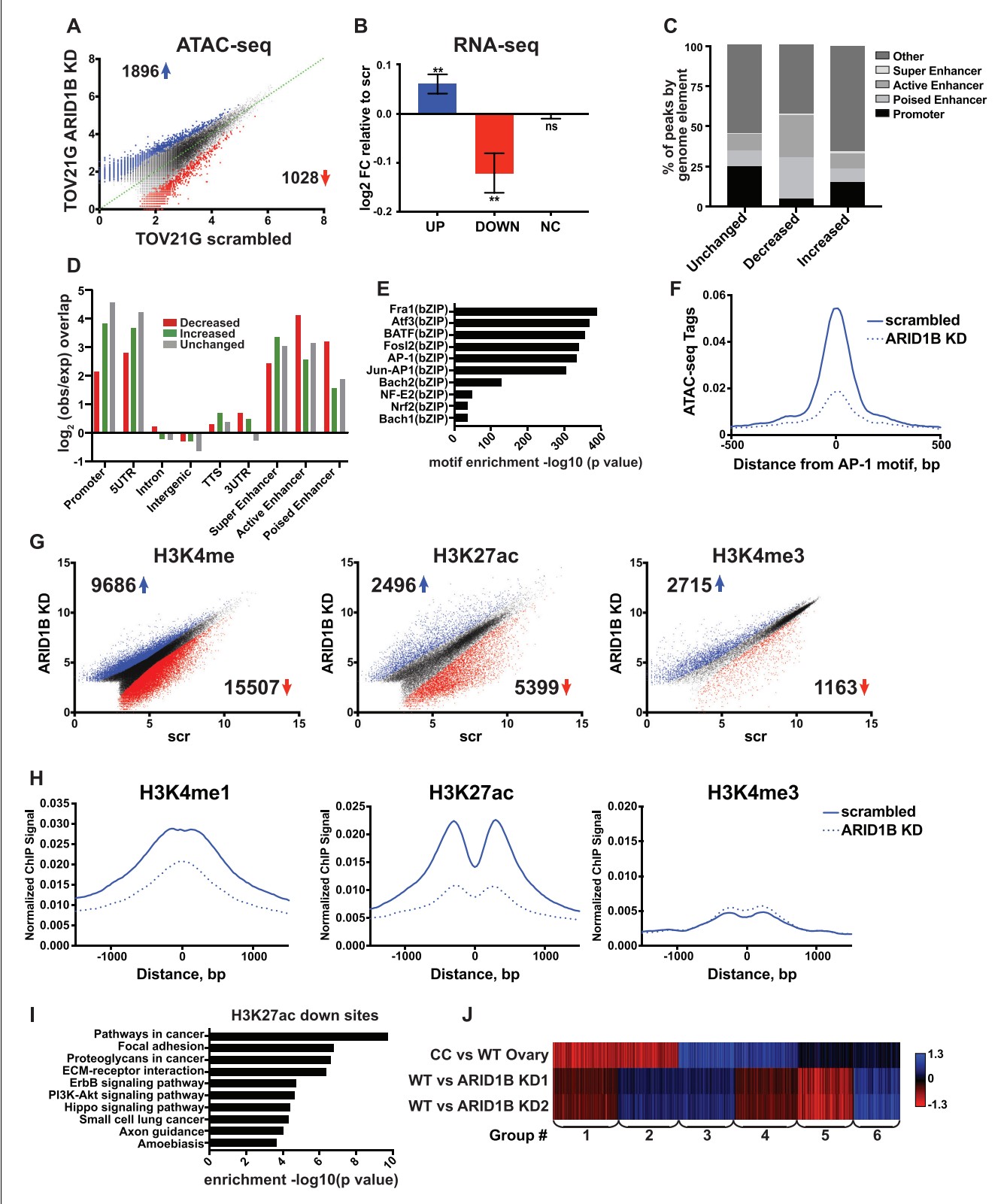

**Figure 7.** ARID1B function at enhancers is conserved in ARID1A mutant ovarian clear cell carcinoma cells. (**A**) ATAC-seq differential peaks in TOV21G cells expressing shRNAs to scrambled control (scrambled) or ARID1B (ARID1B KD). Cell lines were validated for ARID1B expression and function (*Figure 7—figure supplement 1*). Blue and red dots represent differential ATAC-seq peaks whose read density is increased or decreased by 1.5 fold or more, FDR < 0.05, respectively, from two independent biological replicates. Numbers in the upper left and lower right corners refer to numbers of
*Figure 7 continued on next page*

*Figure 7 continued*

called peaks increasing or decreasing in accessibility. (B) Log2 fold change in RNA expression for genes nearest to differentially called accessible sites. Changes are shown for top 10% of genes from increased, decreased, or unchanged (NC) accessible sites for ARID1B KD cells relative to scrambled TOV21G cells. Ns, not significant; **, p<0.01. (C) Percent of unchanged, increased, and decreased accessible sites called from scrambled versus ARID1B KD TOV21G cells present in promoters, poised enhancers, active enhancers, super enhancers, or other genomic elements. (D) Enrichment of unchanged, increased, and decreased accessible sites called from scrambled versus ARID1B KD TOV21G cells at promoters, poised enhancers, active enhancers, super enhancers, 5' and 3' untranslated regions (UTR), introns, intergenic regions, and transcription termination sites (TTS). (E) Motif enrichment in decreased accessible sites from scrambled versus ARID1B KD TOV21G cells. (F) ATAC-seq tag density at AP-1 motifs in scrambled and ARID1B KD TOV21G cells. (G) Differential peak calls from H3K27ac, H3K4me, or H3K4me3 ChIP-seq from TOV21G cells expressing shRNAs to scrambled control (scr) versus shRNAs to ARID1B (ARID1B KD). Blue and red dots represent differential ChIP-seq peaks whose read density increased or decreased by 1.5 fold or more, FDR < 0.05, respectively, from two independent biological replicates. Numbers in the upper left and lower right corners refer to numbers of increased and decreased peaks. (H) Normalized average ChIP-seq tag density for H3K4me, H3K27ac and H3K4me3 from two independent biological replicates at decreased accessible sites from WT and ARID1B KD TOV21G cells. Reads are centered on middle of accessible region ±1.5 kb. Normalized ChIP-seq tag density for H3K4me, H3K27ac and H3K4me3 at increased accessible sites from WT and ARID1B KD TOV21G cells can be found in *Figure 7—figure supplement 2*. (I) Pathway analysis for accessible sites with reduced H3K27ac in ARID1B KD TOV21G cells. (J) *k*-means clustering of 4878 differentially expressed genes in TOV21G cells expressing shRNAs to scrambled control (scrambled) or ARID1B (ARID1B KD1, KD2) from two independent biological replicates compared to expression in ovarian clear cell carcinoma versus normal ovaries (CC vs WT Ovary).

DOI: https://doi.org/10.7554/eLife.30506.027

The following source data and figure supplements are available for figure 7:

**Source data 1.** Proliferation of TOV21G cells with ARID1B knockdown.

DOI: https://doi.org/10.7554/eLife.30506.030

**Figure supplement 1.** Knockdown of ARID1B is synthetically lethal with ARID1A mutation in ovarian clear cell carcinoma.

DOI: https://doi.org/10.7554/eLife.30506.028

**Figure supplement 2.** Normalized average ChIP-seq tag density for H3K4me, H3K27ac, and H3K4me from two independent biological replicates at increased accessible sites from scrambled or ARID1B KD TOV21G cells.

DOI: https://doi.org/10.7554/eLife.30506.029

these modifications do not change in ARID1B KD TOV21G cells (*Figure 7—figure supplement 2*). Pathway analysis of decreased H3K27ac sites revealed an enrichment of ErbB and PI3K-Akt signaling, pathways in cancer, along with ECM-receptor interaction, focal adhesion, and proteoglycans in cancer (*Figure 7I*). The enrichment of genes in pro-proliferative pathways among sites dysregulated by ARID1B loss suggests that ARID1B maintains accessibility and active enhancer architecture at regulatory regions that control OCCC cell proliferation. Indeed, when we compared ARID1B-dependent genes in TOV21G cells to the differential gene expression profile of primary human OCCCs compared to normal ovaries (GSE6008, Gene Expression Omnibus), we found that genes that are highly upregulated in OCCC and downregulated with ARID1B knockdown (Group 3) were enriched for cell cycle terms (*Figure 7J*). Genes that were downregulated in OCCCs but upregulated with ARID1B knockdown (Group 1) contained genes involved in cellular response to stress and EGFR inhibitor resistance, suggesting a potential compensatory response to the effect of ARID1B on growth pathways. Together these data suggest that the role of the SWI/SNF complex in maintenance of accessibility and enhancer marks is conserved in colorectal carcinomas and OCCCs. Repression of OCCC-upregulated genes by ARID1B KD in TOV21G cells, including those involved in cell cycle progression, may contribute to the synthetic lethality observed with ARID1A mutation in this context.

## Discussion

Inactivating *ARID1A* mutations have been detected across a range of diverse cancer types (*Kadoch et al., 2013*), including 9.4% of colorectal carcinomas (*Cancer Genome Atlas Network, 2012*) and up to 60% of OCCCs (*Jones et al., 2010*; *Wiegand et al., 2010*). In the current study, after confirming the synthetic lethal relationship between ARID1A and ARID1B in colorectal carcinoma proliferation, we uncovered the mechanistic basis underlying this phenomenon by examining chromatin accessibility across the genome. Clustering of accessible sites revealed a range of genomic regions with diverse responses to ARID1A and ARID1B loss (*Figure 6*). ARID1A maintains open chromatin, enhancer marks, and gene expression, while ARID1B also modulates these features, but only in the absence of ARID1A. Although the effects of ARID1B knockdown are limited in

comparison, our findings show that the ability of ARID1B to regulate accessibility underpins its absolute requirement for proliferation of ARID1A-deficient colorectal carcinoma cells and mutant OCCCs.

Our results demonstrate that maintenance of chromatin accessibility is central to the tumor suppressor function of ARID1A. Specifically, the effects of ARID1A versus ARID1B loss on the accessible genome (*Figure 1A*) are consistent with a higher frequency of ARID1A mutations in colorectal and ovarian carcinoma (*Cancer Genome Atlas Network, 2012*; *Jones et al., 2010*; *Wiegand et al., 2010*). The underlying cause of ARID1A dominance over ARID1B is not clear. These homologs are similar in molecular weight and protein sequence, with 60% homology and comparable distribution of protein domains (*Patsialou et al., 2005*; *Wilsker et al., 2005*), and do not exhibit differences in their DNA binding ability (*Wilsker et al., 2004*). In HCT116 cells, ARID1A mRNA and protein is more abundant than ARID1B (*Figure 1B* and data not shown), suggesting that there are more SWI/SNF complexes containing ARID1A than ARID1B in the nucleus. The relative abundance of ARID1A may partially account for its dominant role in tumor suppression, although complex composition and/or unique regulatory interactions of ARID1A-containing complexes may also contribute to the selective advantage of ARID1A mutations in human cancer. In contrast, in ARID1A-mutant cells, residual ARID1B-containing complexes ensure that the minimal requirement for SWI/SNF complex remodeling is met at ARID1A-dependent sites. ARID1B thus serves a compensatory role in HCT116 cells, consistent with limited redistribution of SWI/SNF complexes to new sites upon loss of ARID1A (*Mathur et al., 2017*). We did observe increased accessibility and expression in ARID1A$^{-/-}$ cells for a subset of genes reported to gain SWI/SNF binding and H3K27ac by Mathur and colleagues, however the expression of these genes was not ARID1B dependent (*Figure 5—figure supplement 2*). Our data do not address whether ARID1B is retargeted to new sites upon loss of ARID1A in ovarian epithelium, or whether it provides residual, stable binding at predominantly ARID1A-dependent sites. Nevertheless, we find that the role of ARID1B in maintaining accessibility and active histone marks at enhancers in ARID1A-deficient cells is conserved and mechanistically describes its essential role in the ARID1A-deficient setting.

Along with decreases in accessibility across the genome, we found that loss of SWI/SNF subunits results in a significant number of increased accessible sites (*Figure 1A*, *Figure 7A*). SWI/SNF complex recruitment can lead to the direct eviction of Polycomb Repressive Complexes (PRC2), resulting in the loss of H3K27me3 modification and acquisition of accessibility (*Kadoch et al., 2017*; *Stanton et al., 2017*). While we observed that sites with gained SMARCA4/SMARCC1 binding generally overlap with sites that gain accessibility in ARID1A$^{-/-}$ HCT116 cells (*Figure 2C*), there are far fewer new SWI/SNF binding sites than newly accessible regions. In addition, there was no enrichment of H3K27me3 at Cluster 1–3 sites (*Figure 3A*), indicating that eviction of PRC2 complexes cannot fully account for the increased accessibility that we observe. Notably, loss of ARID1A has been shown to release repression mediated by HDACs at gene promoters, which may be an alternative mechanism by which accessibility is gained in ARID1A$^{-/-}$ HCT116 cells (*Chandler et al., 2015*; *Kim et al., 2016*).

Genome-wide binding assays for subunits of the SWI/SNF complex have revealed that SWI/SNF is widely bound to promoters and 5' regulatory regions of actively expressed genes, as well as intronic and distal regulatory elements (*Euskirchen et al., 2011*; *Mathur et al., 2017*; *Morris et al., 2014*; *Raab et al., 2015*). SWI/SNF complexes are often co-bound with other remodeling complexes at these regulatory regions (*de Dieuleveult et al., 2016*; *Morris et al., 2014*). Amidst this backdrop of regulatory complexity, we find that ARID1A and ARID1B preferentially regulate chromatin accessibility at enhancers (*Figure 3A–D*, *Figure 7C and D*). Loss of accessibility likely affects the recruitment of histone readers and writers that deposit H3K27ac and H3K4me active enhancer marks, as direct interactions between modifiers and the SWI/SNF complex have been previously reported (*Alver et al., 2017*; *Huang et al., 2003*; *Nagl et al., 2007*; *Ogiwara et al., 2011*). Accordingly, we observe a reduction in active enhancer marks at ARID1A/1B-dependent accessible sites in colorectal and OCCC lines (*Figure 3F and G*, *Figure 7G and H*). Interestingly, sites with relatively low basal accessibility and typical enhancer marks exhibit greater dependence on ARID1A and ARID1B function (*Figure 1D and E*, *Figure 3A–D*, *Figure 7C and D*), while super enhancer regions, active promoters, and sites with higher basal accessibility are less affected by ARID1A and ARID1B loss. Additional chromatin remodelers, such as PBAF, CHDs, and Mi-2β/NuRD complexes, may maintain chromatin architecture at unaffected sites in the absence of SWI/SNF. In contrast, we find that SWI/

SNF complex binding is most enriched at ARID1A/1B-dependent sites (*Figure 2B* and *Figure 3D*), suggesting that SWI/SNF-mediated regulation of accessibility is correlated with SWI/SNF complex density.

Accessible genomic regions maintained by ARID1A and ARID1B are bound by subunits of the AP-1 complex (*Figure 4A–C*, *Figure 7E and F*). These findings are in agreement with previous reports demonstrating the enrichment of the AP-1 motif at SWI/SNF binding sites (*Mathur et al., 2017*; *Morris et al., 2014*; *Raab et al., 2015*). In vitro, the AP-1 transcription factor complex is known to directly bind and recruit SWI/SNF through interaction with the BAF60A subunit, resulting in transactivation of AP-1 target genes (*Ito et al., 2001*). We observed diminished binding of this transcription factor complex concomitant with diminished mononucleosome spacing at sites with repressed accessibility following ARID1A and ARID1B loss, suggesting that SWI/SNF remodels nucleosomes to facilitate AP-1 binding (*Figure 4G and H*). AP-1 and SWI/SNF could cooperate in chromatin opening, as AP-1 has the ability to potentiate accessibility and prime sites for binding of additional transcription factors (*Biddie et al., 2011*). Together this implies an important relationship between chromatin remodeling and transcription factor complexes in controlling gene expression, one that may involve both eRNA transcription and enhancer-promoter looping. Indeed, ARID1A/1B-dependent sites bound by AP-1 exhibit active transcription (*Figure 5C*) and interactions with TSSs of downregulated genes via RNA Pol II (*Figure 5D*), indicating that SWI/SNF complexes cooperate with transcription factors at distal regulatory sites to impact gene expression.

ARID1A/1B-dependent accessible regions were uncovered within the *MET* gene, consistent with loss of MET expression in ARID1A$^{-/-}$ ARID1B KD HCT116 cells. MET expression moderately enhanced the proliferation of ARID1A$^{-/-}$ ARID1B KD cells, but did not fully rescue the proliferation defect caused by ARID1A/1B loss (*Figure 5J*). Thus, loss of MET expression is a contributing factor, but cannot fully explain the dependence of ARID1A$^{-/-}$ HCT116 cells on ARID1B. Rather, the alteration of multiple compensatory factors likely accounts for the synthetic lethality observed between these SWI/SNF subunits. Indeed, loss of ARID1A/1B profoundly affected the expression of many RTKs and their ligands, as well as downstream signaling adapters in the Ras and PI3K-Akt pathways and transcription factors of the AP-1 family (*Figure 5—figure supplement 3*, *Figure 7I*). Our data suggest that by regulating enhancer accessibility, active enhancer histone modifications, and transcription factor binding, ARID1A/1B maintain expression of genes involved in these pro-proliferative pathways. Given the importance of these pathways in a broad range of cancer types, including colorectal and epithelial ovarian cancer profiled in this study, our data argue for SWI/SNF inhibition as a potential therapeutic approach for ARID1A mutant cancers independent of cancer type. Further, a direct interaction between SWI/SNF and AP-1 could be crucial for initiating and/or maintaining oncogenic signaling, and disruption of binding by specific chemical inhibition represents a potential strategy for inhibiting SWI/SNF function for therapeutic effect.

# Materials and methods

## Key resources table

| Reagent type (species) or resource | Designation | Source or reference | Identifiers | Additional information |
|---|---|---|---|---|
| cell line (*Homo sapiens*) | HCT116 WT | Horizon Discovery | ATCC: CCL-247 | Human colorectal carcinoma cell line |
| cell line (*H. sapiens*) | HCT116 ARID1A$^{-/-}$ | Horizon Discovery | Horizon Discovery: HD104-049 | Human colorectal carcinoma cell line with homozygous ARID1A knockout by knockin of premature stop codon (Q456*) |
| cell line (*H. sapiens*) | TOV21G | ATCC | ATCC: CRL-11730 | Human ovarian clear cell carcinoma cell line |
| transfected construct (*H. sapiens*) | scrambled shRNA | Dharmacon, GE | RHS4346 | scrambled shRNA control |
| transfected construct (*H. sapiens*) | shARID1B#1 | Dharmacon, GE | V2LHS_201002 | shRNA targeting human ARID1B |

*Continued on next page*

*Continued*

| Reagent type (species) or resource | Designation | Source or reference | Identifiers | Additional information |
|---|---|---|---|---|
| transfected construct (*H. sapiens*) | shARID1B#2 | Dharmacon, GE | V3LHS_306691 | shRNA targeting human ARID1B |
| recombinant DNA reagent | pLX304 (Gateway vector) | Addgene; *Yang et al., 2011* | Addgene: #25890 | Gift from David Root |
| recombinant DNA reagent | pDONR223-MET (Gateway donor) | Addgene; *Johannessen et al., 2010* | Addgene: #23889 | Gift from William Hahn and David Root |
| recombinant DNA reagent | pLX-MET | pLX304, pDONR223-MET | | Created by Gateway cloning using pLX304 and pDONR223-MET |
| antibody | anti-ARID1A (mouse monoclonal) | Santa Cruz Biotechnology | sc-32761 | (1:1000) |
| antibody | anti-ARID1B (mouse monoclonal) | Abcam | ab57461 | (1:1000) |
| antibody | anti-TBP (mouse monoclonal) | Thermo Scientific | MA1-21516 | (1:2000) |
| antibody | anti-Brg1 (mouse monoclonal) | Santa Cruz Biotechnology | sc-17796 | (1:1000) |
| antibody | anti-Brm (rabbit polyclonal) | Bethyl Laboratories | A301-015A | (1:1000) |
| antibody | anti-BAF60A (mouse monoclonal) | Santa Cruz Biotechnology | sc-135843 | (1:1000) |
| antibody | anti-BAF57 (rabbit polyclonal) | Bethyl Laboratories | A300-810A | (1:1000) |
| antibody | anti-BAF53A (rabbit polyclonal) | Novus Biologicals | NB100-61628 | (1:1000) |
| antibody | anti-BAF180 (rabbit polyclonal) | Bethyl Laboratories | A301-591A | (1:1000) |
| antibody | anti-MET (rabbit monoclonal) | Cell Signaling Technology | #8198 | (1:1000) |
| antibody | anti-H3K27ac (rabbit polyclonal) | Abcam | ab4729 | |
| antibody | anti-H3K4me (rabbit polyclonal) | Abcam | ab8895 | |
| antibody | anti-H3K4me3 (rabbit monoclonal) | Millipore | #05–745 | |
| antibody | anti-H3K27me3 (rabbit polyclonal) | Active Motif | #39155 | |
| antibody | anti-FRA1 (rabbit polyclonal) | Santa Cruz Biotechnology | sc-183x | |
| antibody | anti-V5 (mouse monoclonal) | Biorad | MCA1360 | (1:2000) |
| antibody | Alexa 488 goat anti-mouse secondary | LI-COR | 926–68070 | (1:20,000) |
| antibody | Alexa 555 goat anti-rabbit secondary | LI-COR | 926–32211 | (1:20,000) |
| sequence-based reagent | *TGFA* F | Eton Bioscience | CTCTACCAGGGCCGAGTTC | |
| sequence-based reagent | *TGFA* R | Eton Bioscience | TCAAGGCCTCGTGTCACAG | |
| sequence-based reagent | *NRIP1* F | Eton Bioscience | GGACACCCAAACCTTCATCC | |

*Continued on next page*

*Continued*

| Reagent type (species) or resource | Designation | Source or reference | Identifiers | Additional information |
|---|---|---|---|---|
| sequence-based reagent | NRIP1 R | Eton Bioscience | CAGTAAGACCCTGGCAGCAT | |
| sequence-based reagent | CALB2 F | Eton Bioscience | ACTGAACTCATCCCACCAGG | |
| sequence-based reagent | CALB2 R | Eton Bioscience | CATTTCCCGTTTCCTGGGTG | |
| sequence-based reagent | PLAU F | Eton Bioscience | GGACCAGCTTTAGTTCCCCT | |
| sequence-based reagent | PLAU R | Eton Bioscience | GGAGGGAGGCAGCATTCTT | |
| sequence-based reagent | MET R1 F | Eton Bioscience | AAGTCACATCTCCAGCGTCC | |
| sequence-based reagent | MET R1 R | Eton Bioscience | CAGGAGTAGCTGAGCCCTTG | |
| sequence-based reagent | MET R2 F | Eton Bioscience | TCAGACATTTGGCACCTCTG | |
| sequence-based reagent | MET R2 R | Eton Bioscience | TCATTTTCCCAATGGTAGCC | |
| sequence-based reagent | Negative region F | Eton Bioscience | GGACAACTCAGGGATGCAAT | |
| sequence-based reagent | Negative region R | Eton Bioscience | GCAGAAGAGAGCCAACCAAC | |
| software, algorithm | HOMER v4.8 | *Heinz et al., 2010* | | http://homer.salk.edu |
| software, algorithm | STAR alignment tool (V2.5) | *Dobin et al., 2013* | | Used within HOMER v4.8 |
| commercial assay or kit | CellTiter-Glo | Promega | G7571 | |
| chemical compound, drug | Tagment DNA Enzyme | Illumina | 15027865 | Part of Nextera DNA Library Prep kit (FC-121–1031, Illumina) |

## Cell culture

HCT116 and HCT116 ARID1A$^{-/-}$ cells were purchased from Horizon Discovery with cell line authentication information provided (STR Profile: Amelogenin: X, Y; CSF1PO: 7, 10; D13S317: 10, 12; D16S539: 11, 13; D5S818: 10, 11; D7S820: 11, 12; THO1: 8, 9; TPOX: 8, 9; vWA: 17,22; ATCC). Both lines were used within six months of receipt. HCT116 ARID1A$^{-/-}$ cells were generated by knock-in of a premature stop codon at Q456 (Horizon Discovery). HCT116 and HCT116 ARID1A$^{-/-}$ cells were grown in RPMI1640 media (Corning CellGro) supplemented with 10% FBS (Omega Scientific, Inc.) and 1% Penicillin/Streptomycin (Life Technologies). TOV21G cells were purchased from ATCC with cell line authentication information provided (STR Profile: Amelogenin: X; CSF1PO: 13, 15; D13S317: 11, 12; D16S539: 10, 12; D5S818: 12, 13; D7S820: 12; THO1: 7, 9.3; TPOX: 8, 11; vWA: 17; ATCC). Cells were used within six months of receipt. TOV21G cells were grown in 1:1 mixture of Medium 199 (Life Technologies) containing 2.2 g/L sodium bicarbonate and MCDB 105 (Life Technologies) containing 1.5 g/L sodium bicarbonate, supplemented with 15% FBS and 1% Penicillin/Streptomycin. All cells were grown at 37°C with 5% $CO_2$. All cell lines were negative for mycoplasma when tested by MycoAlert Mycoplasma Detection Kit (Lonza, Basel, Switzerland)

## shRNA knockdown

Lentiviral GIPZ shRNAs (Dharmacon, GE) targeting ARID1B (shARID1B#1: V2LHS_201002, shARID1B#2: V3LHS_306691) or scrambled control (RHS4346) were used to infect HCT116 and TOV21G cells. Infected cells were selected with puromycin (HCT116: 2 µg/ml; TOV21G: 4 µg/ml) for up to 7 days before performing further experiments.

## Forced expression of MET

MET expression was forced by lentiviral transduction of a blasticidin-resistant plasmid containing MET, generated using the Gateway cloning system with pDONR223-MET (*Johannessen et al., 2010*) and pLX304 (*Yang et al., 2011*). pDONR223-MET was a gift from William Hahn and David Root (Addgene plasmid # 23889). pLX304 was a gift from David Root (Addgene plasmid # 25890).

## CellTiter-Glo assay

Cells were harvested by trypsinisation and plated as single cells at 100 cells/well in 96-well plates. Triplicate wells were plated for each condition and media was replaced every 3 days. For proliferation analysis, 25 µl CellTiter-Glo reagent (Promega) was added to each well and cells incubated for 2 min on orbital shaker before resting for 10 min prior to analysis. Luminescence was then measured using Tecan Infinite M1000 Pro plate reader.

## Antibodies

Primary antibodies used for western blotting are as follows: ARID1A (Santa Cruz Biotechnology, sc-32761), ARID1B (Abcam, ab57461), TBP (Thermo Scientific, MA1-21516), Brg1 (Santa Cruz Biotechnology, sc-17796), Brm (Bethyl Laboratories, A301-015A), BAF60A (Santa Cruz Biotechnology, sc-135843), BAF57 (Bethyl Laboratories, A300-810A), BAF53A (Novus Biologicals, NB100-61628), BAF180 (Bethyl Laboratories, A301-591A), MET (Cell Signaling Technology, #8198), V5 (Biorad, MCA1360). Antibodies used for ChIP: H3K27ac (Abcam, ab4729), H3K4me (Abcam, ab8895), H3K4me3 (Millipore, #05–745), H3K27me3 (Active Motif, #39155), FRA1 (Santa Cruz Biotechnology, sc-183x).

## Immunoblotting

Nuclear fractions were prepared by incubation on ice in buffer A (25 mM HEPES, pH 7.6, 5 mM MgCl2, 25 mM KCl, 0.05 mM EDTA, 10% glycerol, 0.1% NP-40, 1 µM DTT, supplemented with protease inhibitors). After collection by centrifugation, pellets were resuspended in RIPA buffer (50 mM Tris, pH 8.0, 150 mM NaCl, 0.1% SDS, 0.5% sodium deoxycholate, 1% NP-40, 5 mM EDTA, 1 µM DTT, supplemented with protease inhibitors) and incubated on ice for 10 min. Whole cell lysates were prepared by incubation in RIPA buffer on ice for 10 min. Lysates were cleared by centrifugation and supernatants transferred to new tubes before measuring protein concentration by BCA assay. Equal protein amounts were added to loading buffer and samples then loaded onto 8–12% Bis-Tris gels. Following transfer, blots were incubated with primary and secondary antibodies before detection using Odyssey Imaging system (LI-COR Biosciences).

## Glycerol gradient sedimentation

Nuclei were isolated from cells by incubation in buffer A (25 mM HEPES, pH 7.6, 5 mM MgCl2, 25 mM KCl, 0.05 mM EDTA, 10% glycerol, 0.1% NP-40, 1 µM DTT, supplemented with protease inhibitors) on ice, centrifuging 1000xg for 5 min to collect. Nuclei were incubated in buffer C (10 mM HEPES, pH 7.6, 3 mM MgCl2, 100 mM KCl, 0.05 mM EDTA, 10% glycerol, 1 µM DTT, supplemented with protease inhibitors) and ammonium sulfate added to 0.3M final concentration before incubating at 4°C for 30 min. Lysates were cleared by ultracentrifugation at 100,000 rpm (MLA-150 rotor). Supernatants were collected and 0.3 g/ml ammonium sulfate powder added before incubating on ice for 20 min, mixing once after 10 min. Proteins were pelleted by ultracentrifugation as above. Pellets were resuspended in HEMG 0 buffer (25 mM HEPES, pH 7.9, 12.5 mM MgCl2, 100 mM KCl, 0.1 mM EDTA, 1 µM DTT, supplemented with protease inhibitors) and protein amounts quantitated. 10 ml 10–30% Glycerol gradients were prepared as previously described (*Dykhuizen et al., 2013*) and 600 µg protein overlaid before centrifugation in a SW40 swing bucket rotor at 40,000 rpm for 16 hr at 4°C. Twenty 0.5 ml fractions were then collected for immunoblotting analysis.

## ATAC-seq

ATAC-seq was performed as previously described (*Buenrostro et al., 2013*). Briefly, 50,000 cells were washed with cold PBS, collected by centrifugation then resuspended in resuspension buffer (10 mM Tris-HCl, pH 7.4, 10 mM NaCl, 3 mM MgCl2). After collection, cells were lysed in lysis buffer (10 mM Tris-HCl, pH 7.4, 10 mM NaCl, 3 mM MgCl2, 0.1% NP-40) and collected before incubating in

transposition mix containing Tn5 transposase (Illumina). Purified DNA was then ligated with adapters, amplified and size selected for sequencing. Library DNA was sequenced with paired end 42 bp reads.

## Chromatin immunoprecipitation (ChIP)

Cells were harvested and crosslinked in 1% formaldehyde for 10 min before quenching with glycine for 5 min on ice. Cells were pelleted by centrifugation and snap frozen in dry ice before storage at −80°C. Pellets were thawed on ice and resuspended in rinse buffer 1 (50 mM HEPES pH 8.0, 140 mM NaCl, 1 mM EDTA, 10% glycerol, 0.5% NP40, 0.25% Triton X100), collected by centrifugation then resuspended in rinse buffer 2 (10 mM Tris pH 8.0, 1 mM EDTA, 0.5 mM EGTA, 200 mM NaCl). Cells were washed and resuspended in shearing buffer before sonication using Covaris E220 (0.1% SDS, 1 mM EDTA, pH 8, 10 mM Tris HCl, pH 8). For ChIPs using antibodies for histone modifications, $10^6$ cells in 1 ml shearing buffer were added to 1 ml Covaris tubes and sheared for 12 mins at 140W with 10% duty factor. For FRA1 ChIPs, cells were sheared for 8 mins at 140W with 5% duty factor. DNA was then made up to 1x IP buffer (50 mM HEPES/KOH pH 7.5, 150 mM NaCl, 1 mM EDTA, 1% Triton X100, 0.1% DOC, 0.1% SDS, supplemented with protease inhibitors) and 3 μg antibody added for overnight incubation with rolling at 4°C. Antibody bound DNA was recovered using a 1:1 mixture of Protein A and Protein G beads, washed and treated with Proteinase K and RNAse A. Purified ChIP DNA was then used for ChIP-qPCR or library generation for ChIP-seq.

## ChIP-qPCR

Primers used for ChIP-qPCR were as follows: *TGFA*: F: CTCTACCAGGGCCGAGTTC, R: TCAAGGCCTCGTGTCACAG; *NRIP1*: F: GGACACCCAAACCTTCATCC, R: CAGTAAGACCC TGGCAGCAT; *CALB2*: F: ACTGAACTCATCCCACCAGG, R: CATTTCCCGTTTCCTGGGTG; *PLAU*: F: GGACCAGCTTTAGTTCCCCT, R: GGAGGGAGGCAGCATTCTT; *MET* R1: F: AAGTCACATC TCCAGCGTCC R: CAGGAGTAGCTGAGCCCTTG; *MET* R2: F: TCAGACATTTGGCACCTCTG R: TCATTTTCCCAATGGTAGCC; Negative region: F: GGACAACTCAGGGATGCAAT, R: GCAGAAGA-GAGCCAACCAAC. 0.5 μl ChIP DNA was analysed with primers (200 μM) and FastStart Universal SYBR Green Master (Roche) on BioRad CFX Connect real-time PCR machine.

## ChIP-seq library prep

5 ng ChIP DNA was used to prepare libraries using the NuGen Ovation Ultralow Library System V2 following the manufacturer's instructions. Briefly, DNA ends were repaired before ligation of adapters for sequencing. Following bead purification, libraries were amplified by PCR then purified and size-selected for sequencing with single end 50 bp reads.

## RNA isolation and RNA-seq library preparation

RNA from $10^6$ cells was isolated using Quick-RNA Miniprep Kit (Zymo Research). mRNA was then isolated using NEBnext Poly(A) mRNA Magnetic Isolation Module (New England BioLabs) with 5 μg input RNA according to the manufacturer's guidelines. Libraries were prepared for RNA-seq using NEBnext Ultra RNA Library Prep Kit for Illumina following the manufacturer's instructions. mRNA was fragmented and purified before first strand and second strand synthesis. Double-stranded cDNA was then purified and ends repaired before dA tailing and adapter ligation. After cleanup and size selection, cDNA libraries were amplified and purified before sequencing with single end 50 bp reads.

## RNA preparation for eRNA analysis

RNA from $10^6$ cells was isolated using Quick-RNA Miniprep Kit (Zymo Research). cDNA synthesis was performed with Superscript III Reverse Transcriptase (ThermoFisher Scientific) following the manufacturer's protocol with 2.5 μg input RNA and 50 ng/μl random hexamers to enrich for small RNAs. The resulting cDNA was used for RT-PCR analysis with primers used in ChIP-qPCR (see above).

## Data analysis and datasets used

Data were processed using HOMER v4.8 (http://homer.salk.edu/) (*Heinz et al., 2010*). Sequencing metrics for datasets generated in the current study are provided in *Supplementary file 1*. Datasets for DNAse I hypersensitivity (ENCODE: ENCSR000ENM), SMARCA4 and SMARCC1 (*Mathur et al., 2017*), GRO-seq (*Galbraith et al., 2013*), FRA1, JUND, CTCF, ATF3 (ENCODE: ENCSR000BTE, ENCSR000BSA, ENCSR000BSE, ENCSR000BUG), POL2RA ChIA-PET (ENCODE: ENCSR000BZX), human ovarian clear cell and normal ovary microarray (GSE6008, Gene Expression Omnibus) were used.

## ATAC-seq clustering analysis

Paired end 42 bp reads were aligned to hg38 using STAR alignment tool (V2.5) (*Dobin et al., 2013*). ATAC-seq peaks were called using the findPeaks program within HOMER using parameters for DNAse-seq (-style dnase). Peaks were called when enriched >4 fold over local tag counts.

Differential peaks were found using edgeR (*Robinson et al., 2010*) by merging peaks from control and experiment groups and called using getDifferentialPeaks with fold change $\geq 1.5$ or $\leq -1.5$, FDR < 0.05. Peaks were annotated by mapping to nearest TSS using the Homer annotatePeaks tool.

## Clustering analysis

Common peaks shared by two biological replicates in WT, ARID1A$^{-/-}$, and ARID1A$^{-/-}$ ARID1B KD samples were merged and used to generate clusters based on peak shape in WT sample. $k$-means clustering was used to group according to peak tag density changes in ARID1A$^{-/-}$ vs. WT cells. Tag densities from ARID1B KD and ARID1A$^{-/-}$ ARID1B KD samples were then overlaid on these clusters. Heatmaps were created for accessible sites using annotatePeaks with -ghist parameter in HOMER. Window width was set as 1000 bp ($\pm$500 bp). Clustering by $k$-means was performed using Gene Cluster 3.0 (http://bonsai.hgc.jp/~mdehoon/software/cluster/software.htm) using Correlation (centered) for the Similarity Metric. Clusters were visualized using Java Treeview v1.1.6r4.

## Motif analysis

Sequences within 100 bp of peak centers were compared to known motifs in the HOMER database using the findMotifsGenome.pl command with default parameters.

## Nucleosome spacing

We isolated paired-end ATAC-seq fragments between 180 and 247 bp (*Buenrostro et al., 2013*) and plotted the mean dyad density at single nucleotide resolution around AP-1 motifs. The nucleotide of greatest tag density up- and downstream of the motif was found and the spacing between mononucleosomes was determined.

## ChIP-seq analysis

Single-end 50 bp reads were aligned to hg38 using STAR alignment tool (V2.5) (*Dobin et al., 2013*). ChIP-seq peaks were called using the findPeaks program within HOMER using default parameters for histone ChIP-seq (-style histone). Peaks were called when enriched >2 fold over input controls and >4 fold over local tag counts, with FDR < 0.001. ChIP-seq peaks within a 1000 bp range were stitched together to form ChIP-seq regions. Regions were annotated by mapping to nearest TSS using annotatePeaks.pl. Differential regions were found using edgeR by merging peaks from control and experiment groups and called using getDiffExpression.pl with fold change $\geq 1.5$ or $\leq -1.5$, FDR < 0.05.

## Identification of enhancer classes

Enhancer sites were defined as H3K4me + regions that are at least 1 kb away from the nearest annotated TSS or H3K4me3 peak. These sites were divided into poised (H3K27ac-) and active (H3K27ac+) (*Creyghton et al., 2010*). We separated H3K4me1 + H3K27ac + peaks into active enhancers and super-enhancers using a modified approach of the original method (*Whyte et al., 2013*). Specifically, we ranked H3K4me1 + H3K27ac + regions at least 1 kb away from the nearest annotated TSS or H3K4me3 peak by H3K27ac + ChIP seq tag density and used the tangent of the curve to call super-enhancers (*Whyte et al., 2013*).

## RNA-seq analysis

Single-end 50 bp reads were aligned to hg38 using STAR alignment tool (V2.5) (*Dobin et al., 2013*). HOMER analyzeRepeats was used to quantify gene expression normalized for length and sequencing depth (FPKM) across the biological replicates. Differential expression analysis was carried out on raw read counts with the edgeR package (*Robinson et al., 2010*) using independent biological replicates to estimate coefficients of biological variation.

## RNA-seq expression of nearest TSS for annotated accessible sites

ATAC-seq peaks were annotated to the closest TSS and the associated FPKM values were determined. For each ATAC-seq cluster, the top 25% expressed genes were used to calculate $log_2$ fold change compared to WT.

## GRO-seq analysis

We used GRO-seq data (*Galbraith et al., 2013*, GSE38140) from WT HCT116 cells to find putative TSSs. These were then compared to accessible sites in ATAC-seq clusters to determine percentage overlap. Intergenic GRO-seq signal for comparison with ATAC-seq clusters was generated by taking intergenic sites at least 3 kb from promoters and excluding transcription termination sites (TTSs).

## Pathway enrichment analysis

Kyoto Encyclopedia of Genes and Genomes (KEGG) Pathway analysis was performed on annotated peaks using HOMER Gene Ontology (GO) analysis. Gene Set Enrichment Analysis was carried out using Hallmark gene sets using default parameters (*Subramanian et al., 2005*).

## Pairwise comparison with ChIA-PET

We used ChIA-PET interaction data with POL2RA in HCT116 cells (ENCODE Project: ENCSR000BZX). ChIA-PET read pairs were aligned to the hg38 genome using bowtie2. Only interaction read pairs spanning between 5 kb and 1 Mb and containing the same experiment UMI barcode were considered in the analysis. Interaction end-points were assigned to genomic features (i.e. regulated gene TSS, ATAC-seq peaks, etc.) if found within 3 kb of annotated features. Pairwise enrichments between features joined at opposite ends of an interaction were calculated relative to the number of expected interactions based on the randomization of each feature across potential interaction end-points.

## Acknowledgements

Sequencing was carried out by the NGS Core Facility of the Salk Institute with funding from NIH-NCI CCSG: P30 014195, the Chapman Foundation and the Helmsley Charitable Trust. We thank M Ku and T Nguyen for technical support. The Razavi Newman Integrative Genomics and Bioinformatics Core Facility of the Salk Institute is funded through the NIH-NCI CCSG: P30 014195, and the Helmsley Trust. We are grateful to G McVicker (Salk Institute for Biological Studies) for aid with nucleosome spacing analysis. Semi-quantitative protein measurements were carried out by the Mass Spectrometry Core of the Salk Institute with funding from NIH-NCI CCSG: P30 014195 and the Helmsley Center for Genomic Medicine. We thank J Moresco and J Diedrich for technical support. Many thanks to T Hunter (Salk Institute for Biological Studies) and E Dykhuizen (Purdue University) for critical reading of the manuscript.

## Additional information

### Funding

| Funder | Grant reference number | Author |
| --- | --- | --- |
| National Institutes of Health | R00 CA184043-03 | Diana C Hargreaves |
| V Foundation for Cancer Research | V2016-006 | Diana C Hargreaves |
| Genentech Foundation | #G-37246 | Timothy W R Kelso |

The funders had no role in study design, data collection and interpretation, or the decision to submit the work for publication.

## Author contributions

Timothy W R Kelso, Conceptualization, Data curation, Formal analysis, Investigation, Visualization, Methodology, Writing—original draft, Writing—review and editing; Devin K Porter, Data curation, Formal analysis, Investigation, Visualization, Methodology; Maria Luisa Amaral, Formal analysis, Visualization; Maxim N Shokhirev, Resources, Formal analysis, Visualization; Christopher Benner, Resources, Formal analysis, Validation, Visualization; Diana C Hargreaves, Conceptualization, Supervision, Funding acquisition, Investigation, Methodology, Writing—original draft, Project administration, Writing—review and editing

## Author ORCIDs

Diana C Hargreaves (iD) http://orcid.org/0000-0003-3724-3826

## Decision letter and Author response

Decision letter https://doi.org/10.7554/eLife.30506.053
Author response https://doi.org/10.7554/eLife.30506.054

# Additional files

## Supplementary files

• Supplementary file 1: Sequencing metrics for datasets generated in the current study
DOI: https://doi.org/10.7554/eLife.30506.032

• Transparent reporting form
DOI: https://doi.org/10.7554/eLife.30506.033

## Major datasets

The following dataset was generated:

| Author(s) | Year | Dataset title | Dataset URL | Database, license, and accessibility information |
|---|---|---|---|---|
| Kelso TWR, Porter D, Benner C, Hargreaves DC | 2017 | ARID1A and ARID1B loss in HCT116 and TOV21G cells | https://www.ncbi.nlm.nih.gov/geo/query/acc.cgi?acc=GSE101975 | Publicly available at the NCBI Gene Expression Omnibus (accession no. GSE101975) |

The following previously published datasets were used:

| Author(s) | Year | Dataset title | Dataset URL | Database, license, and accessibility information |
|---|---|---|---|---|
| Stamatoyannopoulos J | 2010 | Stam_HCT-116_1 | https://www.ncbi.nlm.nih.gov/geo/query/acc.cgi?acc=GSM736600 | Publicly available at the NCBI Gene Expression Omnibus (accession no. GSM736600) |
| Mathur R, Alver BH, San Roman AK, Wilson BG, et al | 2016 | ARID1A loss impairs enhancer-mediated gene regulation and drives colon cancer in mice | https://www.ncbi.nlm.nih.gov/geo/query/acc.cgi?acc=GSE71514 | Publicly available at the NCBI Gene Expression Omnibus (accession no. GSE71514) |
| Allen MA, Galbraith MD | 2012 | GRO-seq from HCT116 cells | https://www.ncbi.nlm.nih.gov/geo/query/acc.cgi?acc=GSE38140 | Publicly available at the NCBI Gene Expression Omnibus (accession no. GSE38140) |

| | | | | |
|---|---|---|---|---|
| ENCODE DCC | 2012 | HudsonAlpha_ChipSeq_HCT-116_FOSL1_(SC-183)_v042211.1 | https://www.ncbi.nlm.nih.gov/geo/query/acc.cgi?acc=GSM1010756 | Publicly available at the NCBI Gene Expression Omnibus (accession no. GSM1010756) |
| ENCODE DCC | 2012 | HudsonAlpha_ChipSeq_HCT-116_CTCF_(SC-5916)_v042211.1 | https://www.ncbi.nlm.nih.gov/geo/query/acc.cgi?acc=GSM1010903 | Publicly available at the NCBI Gene Expression Omnibus (accession no. GSM1010903) |
| ENCODE DCC | 2012 | HudsonAlpha_ChipSeq_HCT-116_ATF3_v042211.1 | https://www.ncbi.nlm.nih.gov/geo/query/acc.cgi?acc=GSM1010757 | Publicly available at the NCBI Gene Expression Omnibus (accession no. GSM1010757) |
| ENCODE DCC | 2012 | HudsonAlpha_ChipSeq_HCT-116_JunD_v042211.1 | https://www.ncbi.nlm.nih.gov/geo/query/acc.cgi?acc=GSM1010847 | Publicly available at the NCBI Gene Expression Omnibus (accession no. GSM1010847) |
| Wu R, Hendrix N, Kuick R, Misek DE, Hanash SM, Zhai Y, Schwartz DR, Akyol A, Katabushi H, Williams BO, Fearon ER, Cho KR | 2007 | Human ovarian tumors and normal ovaries | https://www.ncbi.nlm.nih.gov/geo/query/acc.cgi?acc=GSE6008 | Publicly available at the NCBI Gene Expression Omnibus (accession no. GSE6008) |
| ENCODE DCC | 2012 | GIS-Ruan_ChiaPet_HCT-116_Pol2 | https://www.ncbi.nlm.nih.gov/geo/query/acc.cgi?acc=GSM970210 | Publicly available at the NCBI Gene Expression Omnibus (accession no. GSM970210) |

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
