## [Decision Letter]

Thank you for submitting your article "Chromatin accessibility underlies synthetic lethality of SWI/SNF subunits in ARID1A-mutant cancers" for consideration by *eLife*. Your article has been favorably evaluated by Jessica Tyler (Senior Editor) and three reviewers, one of whom is a member of our Board of Reviewing Editors. The following individual involved in review of your submission has agreed to reveal their identity: Paul Wade (Reviewer #2).

The reviewers have discussed the reviews with one another and the Reviewing Editor has drafted this decision to help you prepare a revised submission.

Summary:

The manuscript from Kelso and colleagues reports on a genome wide analysis of the binding and function of ARID1A and ARID1B in HCT116 colorectal cancer cell lines. They find a dominant role of ARID1A in influencing chromatin accessibility, features of transcriptional regulatory elements and gene expression, with partial compensation of ARID1A deficiency by ARID1B. AP-1 motifs are highly enriched at sites of ARID1A binding and binding of AP-1 factors is reduced in ARID1 knockout cells. Although investigated in less depth, similar findings are obtained in an ovarian cancer cell line. Analysis of the relatively small number of genes that is altered by combined loss of function of ARID1A and ARID1B provides evidence for a set of growth pathway genes that include MET as being responsible for synthetic lethality in this context.

All three reviewers were of the opinion that the studies were performed at a high level of technical competence and will provide a valuable resource. One reviewer noted, 'The overall strength of this manuscript is the lengthy analysis with multiple datasets that provide a rich picture of roles of ARID1A and ARID1B in the selected system.' The three main concerns raised during discussion were whether the manuscript goes sufficiently beyond the recent studies by Mathur and colleagues, the strength of conclusions regarding MET, and the lack of readily accessible metrics the sequencing assays. With respect to the first concern, there is significant overlap of the findings presented in this manuscript with those of the recent Mathur, et al. paper. However, one reviewer noted, 'This study is unique from previous studies in that it directly analyzed chromatin accessibility in the context of ARID1A and/or ARID1B loss, while previous studies have focused on the changes in histone marks. A second reviewer noted that the findings are complementary and go significantly beyond the Mathur, et al., paper with respect to comparative analysis of the ARID1A and ARID1B proteins and the genome-wide analysis in general. In addition, an important aspect of this manuscript is on the synthetic lethality of combined deletion of ARID1A and 1B, whereas the Mathur, et al., paper primarily focuses on the role of ARID1A as a tumor suppressor gene and studies the consequences of loss of function of ARID1A in mouse colon. However, because of this emphasis, the studies of MET need to be further developed.

Essential revisions:

1) Role of MET in synthetic lethality. This is inferred but not proven and the relatively small difference in expression between ARID1A KO cells and ARID1A KO plus ARID1B knockdown is relatively modest. It may be that small changes of several genes account for the phenotype, but the manuscript would be strengthened by direct experiments. To address this question, the authors should test whether synthetic lethality can be rescued by forced expression of MET. Acceptance of the manuscript is not conditional on the outcome, but this type of experiment is needed to better understand the mechanisms underlying lethality.

2) A table of metrics for all new sequencing experiments is needed to enable an estimation of data quality beyond that provided by the browser tracks. For each RNA-Seq, GRO-Seq and ChIP-seq replicate, the table should include # of reads for each sample, # passing Quality Filter, # uniquely mapped, # after deduplication. For ChIP-seq experiments, the fraction of reads associated with peaks should be provided.

---

## [Author Response]

Essential revisions:1) Role of MET in synthetic lethality. This is inferred but not proven and the relatively small difference in expression between ARID1A KO cells and ARID1A KO plus ARID1B knockdown is relatively modest. It may be that small changes of several genes account for the phenotype, but the manuscript would be strengthened by direct experiments. To address this question, the authors should test whether synthetic lethality can be rescued by forced expression of MET. Acceptance of the manuscript is not conditional on the outcome, but this type of experiment is needed to better understand the mechanisms underlying lethality.

We have now tested forced expression of MET in WT and ARID1A^-/-^ ARID1B KD cells. We find that forced expression of MET did not significantly affect the proliferation of WT cells, although we were unable to overexpress MET above the levels expressed in WT cells (Figure 5—figure supplement 4). However, we were able to rescue MET expression in ARID1A^-/-^ ARID1B KD cells to nearly WT levels. This resulted in a significant albeit small effect on proliferation of HCT116 cells lacking ARID1A and ARID1B (Figure 5). These results demonstrate that loss of MET is a contributing factor, but also show that rescue of this single gene is insufficient to fully counteract the proliferation defect caused by ARID1B loss in ARID1A mutant cells. We show that multiple RTKs are affected in this context, as well as downstream signalling adapters and transcription factors, and thus our data suggest that several genes account for the synthetic lethal phenotype. Indeed, the reduced expression of *PIK3R1*, *JUN, JUND, FOSB*, and *ETS1/2*, all of which are downstream of MET, likely account for why we do not see more of an effect with MET rescue. Changes to the manuscript in the Results and Discussion sections have been made, as well as alteration of Figure 5 and Figure 5—figure supplement 4 to include western blot images for forced MET expression and assays for cell proliferation.

2) A table of metrics for all new sequencing experiments is needed to enable an estimation of data quality beyond that provided by the browser tracks. For each RNA-Seq, GRO-Seq and ChIP-seq replicate, the table should include # of reads for each sample, # passing Quality Filter, # uniquely mapped, # after deduplication. For ChIP-seq experiments, the fraction of reads associated with peaks should be provided.

We created a table of metrics containing this information and have uploaded the table in Excel format as Supplementary file 1.